# Development of a skin- and neuro-attenuated live vaccine for varicella

Wei Wang[1,4], Dequan Pan[1,4], Wenkun Fu[1,4], Xiangzhong Ye[2,4], Jinle Han[2], Lianwei Yang[2], Jizong Jia[2], Jian Liu[1], Rui Zhu[1], Yali Zhang[1], Che Liu[1], Jianghui Ye[1], Anca Selariu[3], Yuqiong Que[1], Qinjian Zhao[1], Ting Wu[1], Yimin Li[2], Jun Zhang [1✉], Tong Cheng [1✉], Hua Zhu [3✉] & Ningshao Xia [1✉]

Varicella caused by the primary infection of varicella-zoster virus (VZV) exerts a considerable disease burden globally. Current varicella vaccines consisting of the live-attenuated vOka strain of VZV are generally safe and effective. However, vOka retains full neurovirulence and can establish latency and reactivate to cause herpes zoster in vaccine recipients, raising safety concerns. Here, we rationally design a live-attenuated varicella vaccine candidate, v7D. This virus replicates like wild-type virus in MRC-5 fibroblasts and human PBMCs, the carrier for VZV dissemination, but is severely impaired for infection of human skin and neuronal cells. Meanwhile, v7D shows immunogenicity comparable to vOka both in vitro and in multiple small animal species. Finally, v7D is proven well-tolerated and immunogenic in nonhuman primates. Our preclinical data suggest that v7D is a promising candidate as a safer live varicella vaccine with reduced risk of vaccine-related complications, and could inform the design of other herpes virus vaccines.

[1] State Key Laboratory of Molecular Vaccinology and Molecular Diagnostics, National Institute of Diagnostics and Vaccine Development in Infectious Diseases, School of Life Sciences, School of Public Health, Xiamen University, Xiamen 361102 Fujian, China. [2] Beijing Wantai Biological Pharmacy Enterprise Co., Ltd., Beijing 102206, China. [3] Department of Microbiology and Molecular Genetics, New Jersey Medical School, Rutgers University, 225 Warren Street, Newark, NJ 070101, USA. [4] These authors contributed equally: Wei Wang, Dequan Pan, Wenkun Fu, Xiangzhong Ye. ✉email: zhangj@xmu.edu.cn; tcheng@xmu.edu.cn; zhuhu@njms.rutgers.edu; nsxia@xmu.edu.cn

Varicella, caused by primary infection with varicella-zoster virus (VZV), is a common disease in infancy and childhood. It was estimated that a minimum of 140 million varicella cases, 4.2 million severe varicella complications leading to hospitalization, and 4200 varicella-related deaths occur globally each year[1]. Although generally self-limiting, varicella can be associated with a variety of severe and life-threatening complications in both immunocompetent and immunocompromised individuals[2,3]. Moreover, after primary varicella infection, VZV establishes latency in the nerve ganglia and may reactivate to cause herpes zoster (HZ)[4]. The reactivation is sometimes complicated by post-herpetic neuralgia (PHN), a severe pain that can persist for years and substantially affect patients' quality of life[4].

The varicella vaccine was first approved for use in routine vaccination for children in the USA in 1995 and has been subsequently introduced into childhood immunization programs in many other countries[5]. In recent decades, the current varicella vaccine has proven to be generally well-tolerated and highly effective in preventing varicella-related morbidity and mortality[6–8]. WHO recommends that varicella vaccination be introduced into routine immunization programs in countries where varicella poses a significant public health burden, and the vaccine coverage should be maintained above 80%[1].

The most widely used varicella vaccines consist of the Oka strain of live-attenuated VZV (vOka). Takahashi's group developed the vOka strain in the 1970s by serially passaging the wild-type parent Oka (pOka) strain of VZV in human and guinea pig cell lines. However, the original vOka virus stock was never subcloned, and thus the current vOka vaccine is a mixture of haplotypes with its attenuation mechanism remaining ill-defined[9,10]. Concerns have been raised about the use of live vOka varicella vaccines. Firstly, vOka-mediated adverse events, a particularly varicella-like rash with a potential of environmental release, have been reported in up to 5% of vaccinated healthy children[11]. Furthermore, vOka retains wild-type neurovirulence and can establish latency in some vaccine recipients and reactivate to cause HZ and neurological complications like meningitis[6,7,12–19]. By deep sequencing, several wild-type single nucleotide polymorphisms (SNPs) in the genomes of vOka viruses have been found to be linked with vaccine-associated varicella and HZ, indicating the existence of less attenuated vOka haplotypes in skin- and neuro-tropism[10]. No one vOka haplotype has been confirmed responsible for vaccine-associated diseases to date[10,20]. However, dormant vOka viruses in human neurons could pose one of the potential risk factors for HZ. Although recent studies have reported lower HZ incidences in vaccinated children than in unvaccinated ones (<18 years old)[14,21,22], it will be decades before evidence accrues to show whether the vOka varicella vaccination would lead to an increase in the incidence of vOka-mediated HZ as vaccine recipients age and their immunity declines. In addition, there is a growing concern that life-long latency and reactivation of herpesviruses in human neurons may contribute to diseases affecting memory and cognitive functions[23–26]. Therefore, a safer next-generation varicella vaccine without or with low pathogenic potential is desirable to improve the acceptance of vaccination against varicella and reduce vaccine hesitancy.

Current advances in molecular virology and the development of reverse genetic systems have led to the identification of many viral genes associated with virulence and tissue tropism of VZV, thus providing a sound basis for the rational design of novel live-attenuated VZV vaccine candidate[27–35]. Notably, the ORF7 gene is the only full-length VZV gene identified to date that is required for virulence in both human skin and neuronal cells, besides being non-essential for viral replication in various cell cultures[30–32]. Thus, a pure and genetically defined VZV mutant that is defective in ORF7 expression could be both skin- and

neuro-attenuated, while retaining its potency to induce protective immunity in children. Here, we reported the construction and characterization of an ORF7-deficient, pOka-derived vaccine strain of VZV, described hereafter as v7D (vaccine strain VZV-7D). The findings from this study have paved the way for a first-in-human clinical trial of v7D vaccine candidate in China (ChiCTR1900022284).

## Results

**Construction of the v7D vaccine candidate**. VZV wild-type pOka strain was used as the parent virus to construct an infectious bacterial artificial chromosome (BAC; b) clone of a recombinant Oka virus (rOka) expressing green fluorescent protein (GFP), designated as brOka-GFP, as described previously[36]. This clone was then modified by mutating the 11-bp region downstream of the ATG start codon of ORF7 into a three-frame stop-codon cassette to create an ORF7-deficient VZV-BAC clone, b7D-GFP (Supplementary Fig. 1a, b). The loxP-flanked BAC vector containing the GFP expression cassette was excised from the viral genomes by co-transfecting b7D-GFP with a Cre expression vector into MRC-5 human embryonic lung fibroblasts (Supplementary Fig. 1c, d), thus reconstituting the ORF7-deficient candidate vaccine virus, v7D. Next-generation sequencing (NGS) analysis was performed on whole-genome sequences of v7D at passage (P) 12, P15, P16, and P25 to determine its genetic stability. Compared with the genomic sequence of VZV-pOka, there was no difference in the sequences of coding regions except for the stop-codon mutation in ORF7 though some mutations were found in non-coding regions of the genomes of v7D from different passages (Supplementary Table 1), suggesting that v7D genome is highly stable. To further prove that the stop-codon mutation, rather than possible mutations in the viral genome, was responsible for any growth defect of the ORF7-deficient VZV mutant, a BAC-derived ORF7-rescued VZV clone, b7R-GFP, was also constructed (Supplementary Fig. 1a, b). The rescue virus 7R and the wild-type virus rOka were recovered from their respective BAC clone and used as controls in subsequent experiments.

**Characterization of v7D replication in cell culture**. At a multiplicity of infection (MOI) of 0.01, the growth kinetics of v7D were comparable to those of rOka, 7R, and the vOka vaccine strain in MRC-5 cells (Fig. 1a). The levels of 12 VZV structural proteins, including capsid proteins (pORF23 and pORF40), tegument proteins (pORF7, pORF9, pORF47, pORF62, and pORF63), and glycoproteins (gE, gB, gH, gI, and gN) were examined by western blotting at two and four days post-infection (dpi) among different infections. While in the absence of ORF7, v7D yielded abundant viral proteins similar to those yielded by rOka, 7R, and vOka in the infected MRC-5 cells (Fig. 1b and Supplementary Fig. 2) and could thus provide key viral antigens as a vaccine candidate.

The replication of v7D was also tested in differentiated SH-SY5Y human neuroblastoma cells (dSY5Y; MOI = 0.01), primary human dermal fibroblasts (HDFs; MOI = 0.01) and primary human embryonic keratinocytes (HEKs; less susceptible to VZV infection;[37,38] infected at an MOI of 0.2 as previously reported[37]). In contrast to the results obtained with rOka, 7R, and vOka, the infectious progeny virus production and viral translation of v7D were markedly impaired in the human skin and neuronal cells (Fig. 1a, b and Supplementary Fig. 2). These correlated with signs of cell damage, including the cleavage of caspase-3 and poly (adenosine diphosphate-ribose) polymerase (PARP), as well as decreases in cell viability in the infected cells (Fig. 1b, c). Therefore, v7D induced little cytopathic effect on these cells compared to those infected with rOka, 7R, and vOka, suggesting

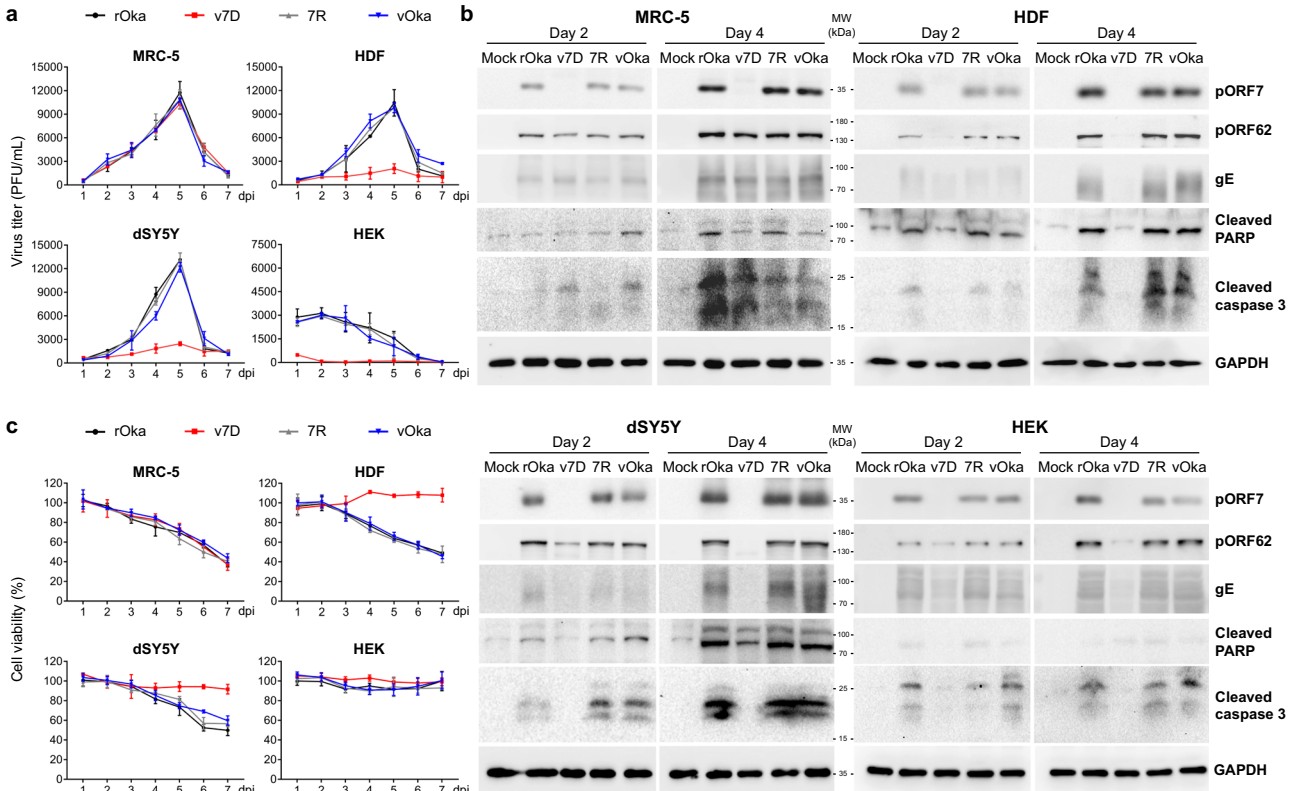

**Fig. 1 v7D is severely attenuated in human skin and neuronal cells in vitro.** MRC-5 cells, dSY5Y cells, HDFs, and HEKs were mock-infected or infected with cell-free rOka, v7D, 7R, and vOka, respectively. For MRC-5 cells, dSY5Y cells and HDFs, MOI = 0.01. For HEKs, MOI = 0.2. **a** Infected cells were harvested daily over a period of 1 week and viral titers were determined by plaque assay in MRC-5 cells to generate growth curves; **b** Lysates were prepared from the harvested cells at 2 and 4 dpi and tested by western blot for markers of viral translation (exemplified by viral proteins pORF7, pORF62 and gE, more in Supplementary Fig. 2) and host cell damage/death (PARP and caspase-3 cleavage); **c** The cell survival was measured by CCK-8 assay at indicated time points after infection. The results are represented as averages ± the SD ($n = 3$ per group). The experiments were repeated twice, and representative results are shown. Source data are provided as a Source Data file.

that v7D was severely attenuated in human skin and neuronal cells in vitro.

In addition, since human peripheral blood mononuclear cells (PBMCs) are preferential targets of VZV infection that play a critical role in disseminating the virus throughout the host to produce clinical diseases and stimulate immune responses[39], v7D infection in PBMCs was evaluated. PBMCs obtained from healthy donors ($n = 6$) were infected with cell-free viruses of rOka, v7D, and vOka (MOI = 0.01), respectively, and analyzed at 3 dpi for viral gene expression and genome replication. First, using flow cytometry analysis, PBMCs infected with v7D showed similar cell-surface expression levels of VZV gE compared with those infected with rOka and vOka (Fig. 2a, b). Meanwhile, using immunofluorescence assay, all VZV-infected PBMCs expressed nuclear pORF62, and perinuclear pORF7 staining was only positive in rOka- and vOka-infected PBMCs but not in v7D-infected ones (Fig. 2c). Next, the viral genome synthesis and gene transcripts (ORF31 and ORF62) were investigated in the infected PBMCs and found to be comparable among rOka, v7D, and vOka infections (Fig. 2d, e). These data suggest that v7D retained the tropism for human PBMCs of wild-type VZV, which could make it competent to be a live-attenuated vaccine in humans.

**In vivo assessment of v7D attenuation in animal models of VZV infection.** VZV is a highly human-specific virus that has little to no capacity of infecting other species, and to date, no natural animal model of VZV infection is available[27]. Early

attempts to establish VZV infection in guinea pigs and cotton rats by experimental inoculation have resulted in the detection of VZV DNA and RNA in the ganglia, which resemble those seen in latently infected human ganglia[40]. Therefore, firstly, the neurotropism of v7D in the guinea pig and cotton rat models was evaluated according to the previous reports. Animals ($n = 6$ per group for both guinea pigs and cotton rats) were inoculated intramuscularly along both sides of the thoracic and lumbar spine with $3 \times 10^5$ plaque-forming units (PFU) of cell-associated viruses of v7D, rOka, and vOka, respectively. After 1 month, animals were euthanized and dorsal root ganglia (DRG) were examined for VZV DNA and RNA by quantitative PCR using probes for ORF31 and ORF62. The results showed that VZV DNA genome and gene transcripts were detectable in ganglia specimens from 2 to 4 of the six animals in groups inoculated with rOka and vOka (Supplementary Fig. 3). In contrast, ganglia from the v7D-inoculated and control animals were negative for VZV DNA and gene transcripts (Supplementary Fig. 3), suggesting the relative impairment in v7D's ability to establish neuronal latency in guinea pigs and cotton rats.

To further confirm the attenuation of v7D in human skin and neuronal cells in a more physiologically relevant system, the previously established SCID-hu mouse models carrying xenografts of human skin or DRG were used as previously reported by the Arvin's lab with certain modifications[41,42]. Experiments were conducted using fetal skin and DRG tissues obtained from healthy donors ($n = 3$). Different from the guinea pig and cotton rat models, the virus inoculum is directly injected into the human

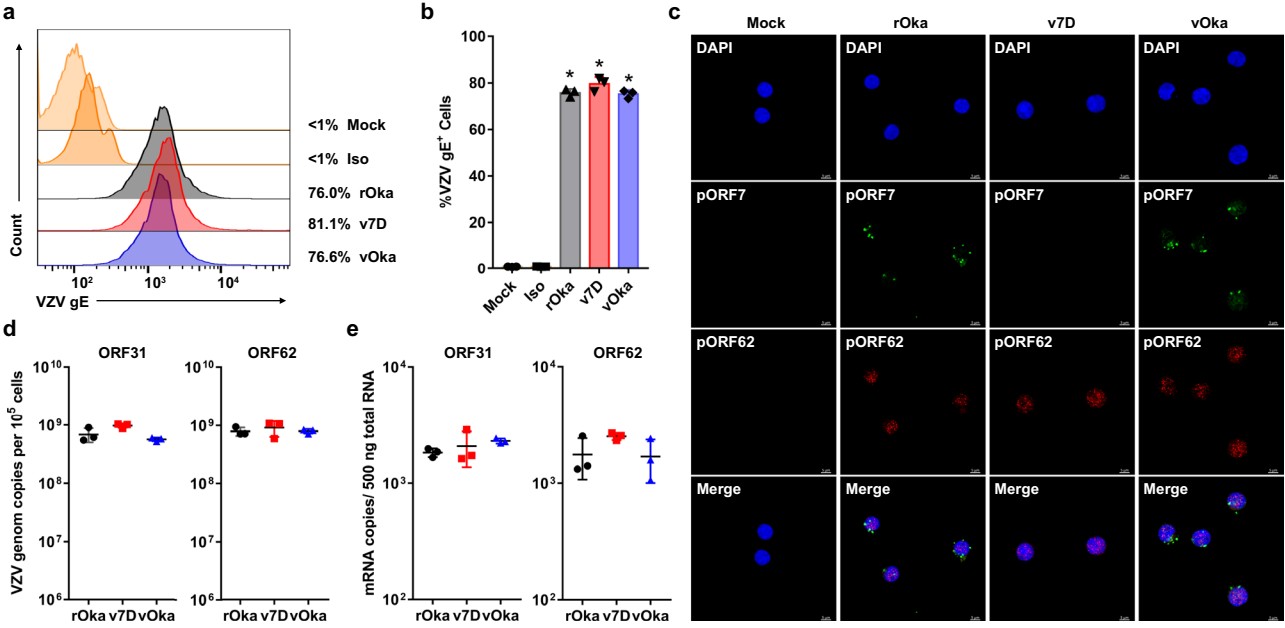

**Fig. 2 v7D retains full infectivity for human PBMCs in vitro.** Human PBMCs were mock-infected or infected with rOka, v7D, and vOka (MOI = 0.01), respectively. The cells were harvested at 3 dpi. **a**, **b** Representative flow cytometry analysis and summary of the cell-surface expression of VZV gE on PBMCs. rOka-infected PBMCs were stained with an isotype antibody (iso) as a control. Numbers on the right side of each graph in (**a**) represent the percent of VZV-gE-positive cells. Asterisks in (**b**) denote a significant difference ($P < 0.05$) compared to the mock-infected controls as determined by one-way ANOVA with Tukey's post hoc test. Detailed information about the statistics and values are provided in Supplementary Table 3. **c** Immunofluorescence analysis of the expression of VZV pORF7 and pORF62 in infected PBMCs. The experiment was repeated twice with similar results, and representative images are shown. Scale bar, 5 μm. **d** Determination of VZV genome copy numbers in infected PBMCs using probes for ORF31 and ORF62. The adjustment was made for ORF62 copy numbers to reflect that the VZV genome contains two copies of ORF62. **e** mRNA transcripts of VZV ORF31 and ORF62 in infected PBMCs. The results are represented as averages ± the SD ($n = 3$ per group). Source data are provided as a Source Data file.

tissues most sensitive to VZV infection in the SCID-hu mouse models. Thus, cell-free viruses instead of cell-associated viruses were used to minimize the influence of pre-existing viral nucleic acids and proteins on experimental outcomes.

First, for the SCID-hu skin mouse model, skin xenografts (~1 cm width × 1 cm length) were inoculated with cell-free viruses of $1 \times 10^4$ PFU (100 μL, 27-gauge needle) of rOka, v7D, 7R, and vOka, respectively. The results showed that inoculation with rOka and 7R caused large cutaneous lesions with disruption of the keratinized layer of the skin and virus spreading into the dermis by day 21, while vOka infection progressed more slowly and resulted in smaller lesions. In contrast, v7D exhibited a severely skin-attenuated phenotype with no lesions found in all inoculated skin xenografts (Fig. 3a). When the infectious virus yields from skin xenografts were compared at 10 and 21 dpi, the mean titers of vOka were lower than those of rOka and 7R at both time points (Fig. 3b). The identity of the recovered vOka viruses was verified by a previously established PCR–RFLP assay[43], compared with wild-type rOka (Supplementary Fig. 4a). As expected from our previous study[30], no infectious virus was recovered from any v7D-inoculated skin xenograft (Fig. 3b).

Second, for the SCID-hu DRG mouse model, DRG xenografts (~1 mm³) were inoculated with cell-free viruses of $1 \times 10^3$ PFU (10 μL, 30-gauge needle) of each virus. When the infectious virus production from DRG xenografts was assessed, infectious virus was recovered from 5/5, 3/5 and 3/5 DRG xenografts inoculated with rOka, 7R, and vOka, respectively, at 14 dpi. The identity of the recovered vOka viruses was verified by PCR–RFLP as above (Supplementary Fig. 4b). No infectious virus was recovered from any of the DRG xenografts inoculated with rOka, 7R, or vOka at 28 and 56 dpi, and v7D did not yield infectious virus in DRG xenografts at any time point. Using immunohistologic analysis,

by day 14, rOka-, 7R-, and vOka-inoculated DRG xenografts exhibited abundant viral protein expression (exemplified by gE) and disruption in tissue architecture, whereas no viral translation and cytopathic changes were observed in v7D-inoculated DRG xenografts (Fig. 3a). The viral genome synthesis and gene transcripts (ORF31 and ORF62) in the inoculated DRGs were also compared at 14, 28, and 56 dpi as described previously[41]. The results showed that rOka-, 7R-, and vOka-inoculated DRGs developed high VZV genome copy numbers and high levels of ORF31 and ORF62 transcripts at 14 dpi, which markedly declined at 28 and 56 dpi as expected (Fig. 3c, d). In contrast, v7D-inoculated DRGs collected at different time points showed undetectable VZV genome copy numbers and gene transcripts (Fig. 3c, d). These data suggest that v7D failed to establish an initial lytic infection and virus persistence in DRGs in vivo, compared with rOka, 7R, and vOka.

**Immunogenicity evaluation of v7D using human dendritic cell-based assays.** Dendritic cells (DCs) are essential antigen-presenting cells (APCs) that act at the first step in initiating adaptive immune responses against pathogens following vaccination or natural infection[44,45]. Therefore, prior to animal testing, the immunogenicity of v7D was compared to that of vOka using human DC-based in vitro assays. First, v7D or vOka was incubated with human CD14+ monocyte-derived immature DCs (iDCs) (MOI = 0.01) over a period of five days, and DC activation phenotypes were assessed using flow cytometry and multiplex cytokine/chemokine array analysis. The results showed that DCs were permissive to VZV infection (Fig. 4a, b) and exposure to both v7D and vOka (3 dpi) significantly increased the percentage of DCs expressing CD40, CD80, CD83, and CD86 costimulatory markers, indicating DC maturation

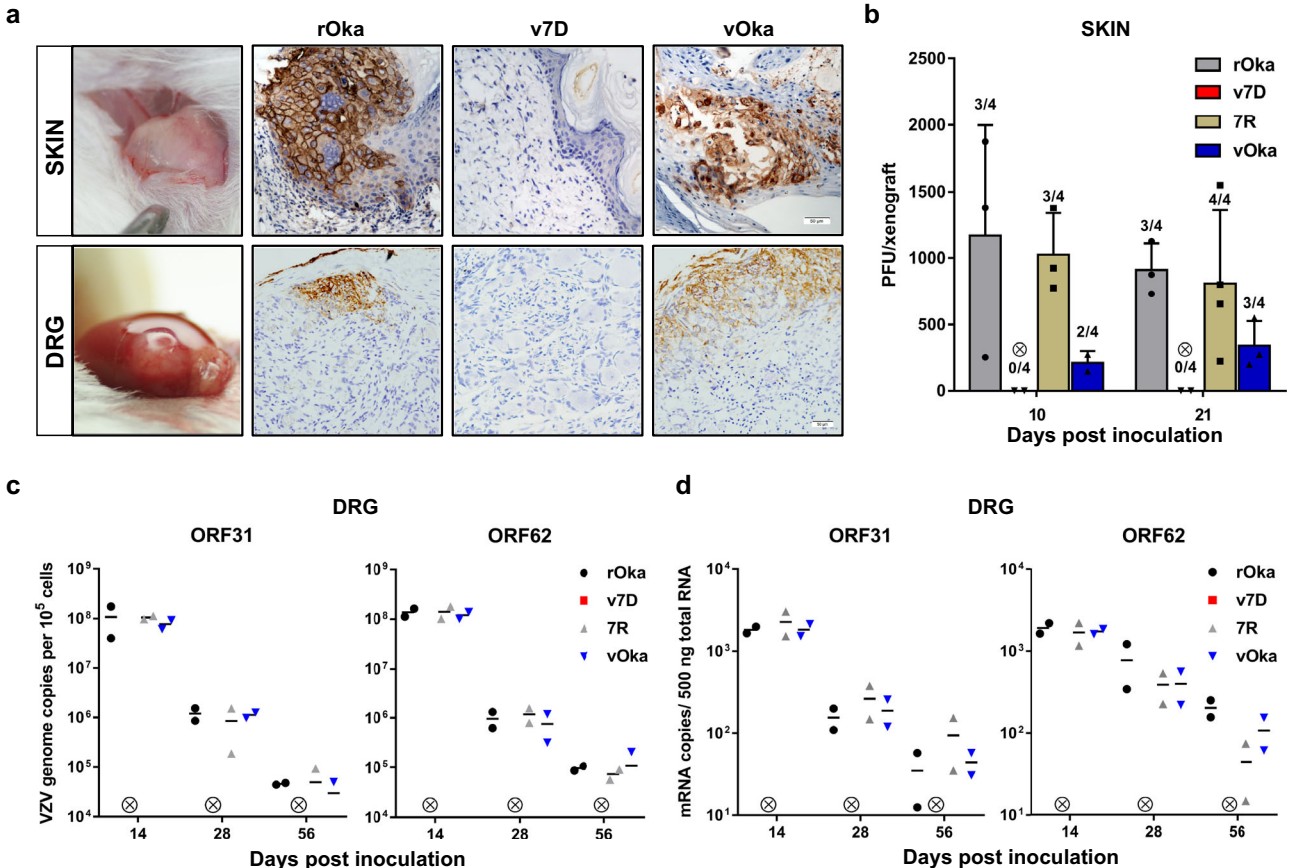

**Fig. 3 v7D is severely attenuated in human skin and neuronal tissues in vivo.** Human fetal skin and DRG tissues engrafted in the SCID-hu mice were mock-infected or infected with cell-free rOka, v7D, 7R, and vOka, respectively. The inoculum doses for skin and DRG were $1 \times 10^4$ and $1 \times 10^3$ PFU per xenograft, respectively. **a** Anti-gE IHC analysis of infected skin and DRG xenografts. Images of infected skin and neuronal cells were captured at 21 and 14 dpi, respectively. Scale bar, 50 μm. **b** Viral titers in skin xenografts were determined by plaque assay in MRC-5 cells at 10 and 21 dpi. The number of xenografts from which infectious virus was recovered per number of inoculated xenografts is shown above each bar. Mean titers per group were calculated based on data from xenografts that yielded viable virus. The results are represented as averages ± the SD. **c** Mean VZV genome copy numbers in infected DRG xenografts ($n = 2$ per group) using probes for ORF31 and ORF62 at 14, 28, and 56 dpi. Adjustment was made for ORF62 copy numbers to reflect that each VZV genome contains two copies of ORF62. **d** mRNA transcripts of VZV ORF31 and ORF62 in infected DRG xenografts ($n = 2$ per group) at 14, 28, and 56 dpi. A circle containing a cross indicates that none of the animals in the v7D group had detectable viral titers, VZV DNA or RNA. The experiments were repeated twice, and representative results are shown. Source data are provided as a Source Data file.

(Fig. 4c and Supplementary Fig. 5). Furthermore, interactions of DCs with both v7D and vOka induced sustained production of similar levels of pro-inflammatory cytokines, including TNF-α and IL-6, as well as chemokines IP-10, MIP-1β, MCP-1, and RANTES without causing cell lysis (Fig. 4d, e), which could aid in DC functions.

Second, to further test the downstream effects of v7D- and vOka-induced DC activation, v7D- or vOka-pulsed iDCs (5 dpi) were co-incubated with autologous CD4+ or CD8+ T cells. Over a 5-day co-culture period, T cells were harvested for proliferative assays and an interferon-γ (IFN-γ) enzyme-linked immunosorbent spot (ELISPOT) assay, and the culture supernatants were also collected for analysis of cytokine/chemokine production. The results showed that co-culture with v7D- and vOka-pulsed iDCs induced comparable proliferative responses (Fig. 4f–h) and secretion of similar levels of Th1/Th2 cytokines (TNF-α, IFN-γ, IL-2, IL-4, IL-6, IL-8, IL-10, IL-13, etc.) and chemokines (IP-10, MIP-1β, MCP-1, and RANTES) (Fig. 4i and Supplementary Fig. 6) in both CD4+ and CD8+ T cells. Together, compared with vOka, these findings define a similar ability of v7D in mediating DC functional activation, which could facilitate subsequent priming of adaptive immunities against VZV infection.

**Immunogenicity evaluation of v7D in small animals.** There is still a lack of ideal animal models that recapitulate the immune response to VZV in humans due to its strict human specificity. In this context, we used small animals most commonly used in preclinical vaccine studies, including mice, rats, rabbits, and guinea pigs, to test the in vivo immunogenicity of v7D in comparison to that of vOka. Animals were immunized subcutaneously with v7D or vOka (500 PFU/dose for mice; 1000 PFU/dose for rats; 1000 PFU/dose for guinea pigs; 2000 PFU/dose for rabbits) at weeks 0, 3, and 6. First, serum samples were collected at 3-week intervals over a 42-week period for evaluation of antibody responses. The titers of anti-VZV IgG antibodies were determined by indirect enzyme-linked immunosorbent assay (ELISA), and the neutralizing antibody titers were determined by plaque reduction neutralization test in MRC-5 cells. The results showed that v7D and vOka at the same doses induced comparable kinetics of anti-VZV IgG and neutralizing antibodies in these animals (Fig. 5a, b). The geometric mean titers (GMTs) peaked 3 weeks after the third vaccination and pairwise comparisons revealed no significant difference between v7D and vOka. In addition, the IgG subclasses distributions were evaluated by ELISA in sera of mice and rats 3 weeks after the first (week 3) and third

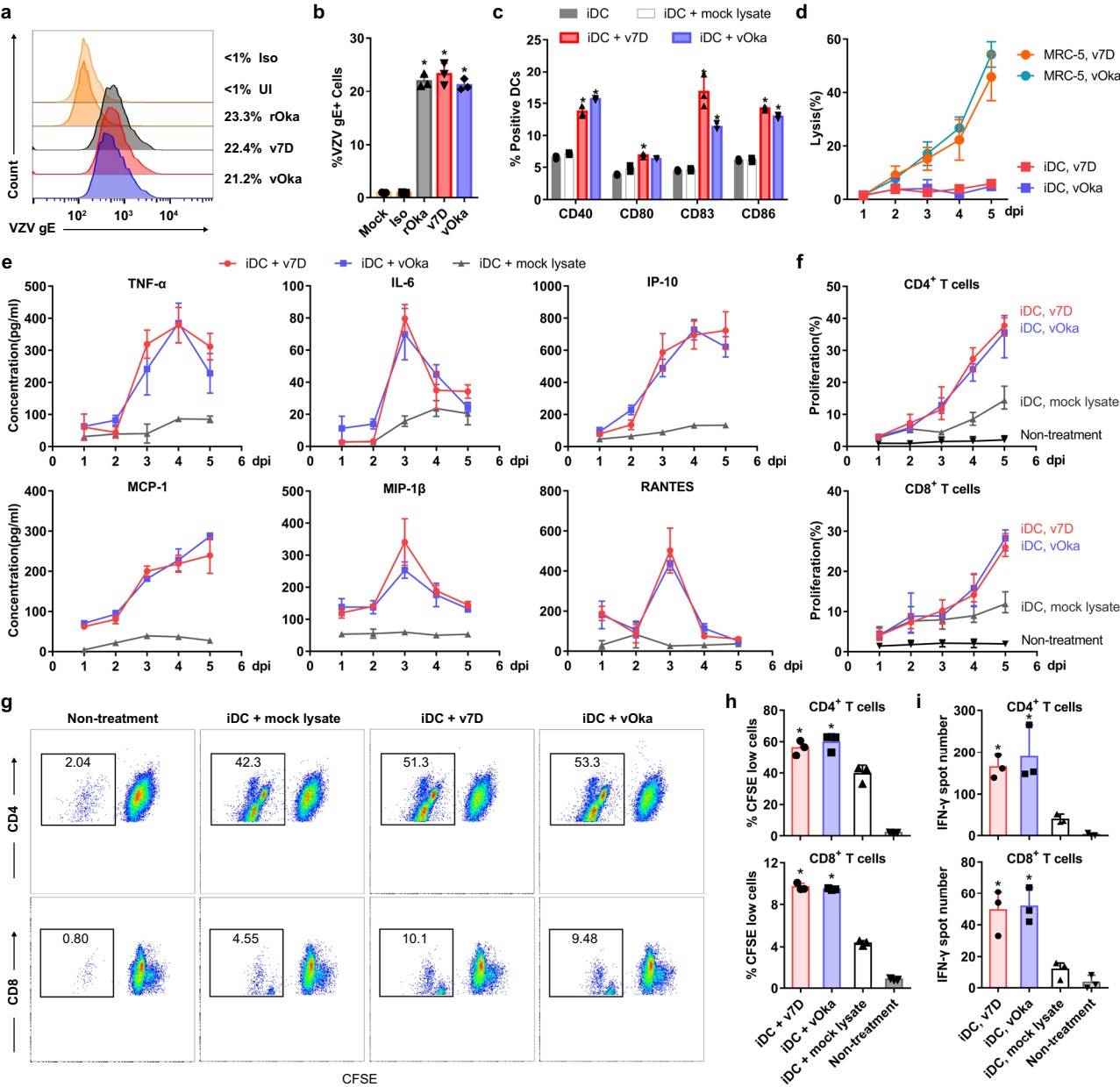

**Fig. 4 v7D induces functional activation of human DCs in vitro.** Human iDCs were generated from CD14[+] monocytes purified from PBMCs, and were treated with rOka, v7D, vOka (MOI = 0.01) or mock-infected MRC-5 cell lysate as a control. **a**, **b** Representative flow cytometry data and summary of VZV-gE expression on iDCs at 3 dpi. rOka-infected iDCs were stained with an isotype antibody (iso) as a control. Numbers on the right side of each graph in (**a**) represent the percent of VZV-gE-positive cells. **c** Flow cytometry analysis of expression of the cell-surface costimulatory markers CD40, CD80, CD83 and CD86 on DCs at 3 dpi. Representative flow cytometry data are shown in Supplementary Fig. 5. **d** Analysis of percent lysis of DCs compared to that of MRC-5 cells after inoculation with v7D and vOka (MOI = 0.01) over five days. **e** Cytokine/chemokine analysis of DC culture supernatants. **f** Analysis of cell proliferation rates of the purified autologous CD4[+] and CD8[+] T cells after co-culture with antigen-pulsed DCs using the BrdU ELISA assay. **g**, **h** Representative flow cytometry data and summary data of CFSE dilution among CD4[+] and CD8[+] T cells after 5-day co-culture with antigen-pulsed DCs. **i** Analysis of IFN-γ production in CD4[+] and CD8[+] T cells after 5 days of co-culture with antigen-pulsed DCs using an ELISPOT assay. The results are represented as averages ± the SD (n = 3 per group). Asterisks denote a significant difference (P < 0.05) compared to the mock controls as determined by one-way (**b**, **h**, **i**) or two-way (**c**) ANOVA with Tukey's post hoc test. Detailed information about the statistics and values are provided in Supplementary Table 3. The experiments were repeated twice using cells from six donors, and representative results are shown. Source data are provided as a Source Data file.

immunization (week 9). The results showed that v7D and vOka induced similar IgG subclass distributions (Fig. 5c, d). In mice, IgG1 was predominant, with slightly elevated IgG2a and IgG2b titers, while in rats, IgG2a and IgG2b were the major subclasses, with slightly elevated IgG1 and IgG2c titers. According to previous studies on IgG subclasses[46–49], the elevated VZV-specific IgG

subclasses indicated that both v7D and vOka induced a mixed Th1/Th2 response with Th2 dominance in mice and rats.

Second, to characterize the cell-mediated immune responses to v7D and vOka, splenocytes were collected from immunized mice and rats 3 weeks after final immunization (week 9) for in vitro cytokine stimulation assays. In response to rOka virus

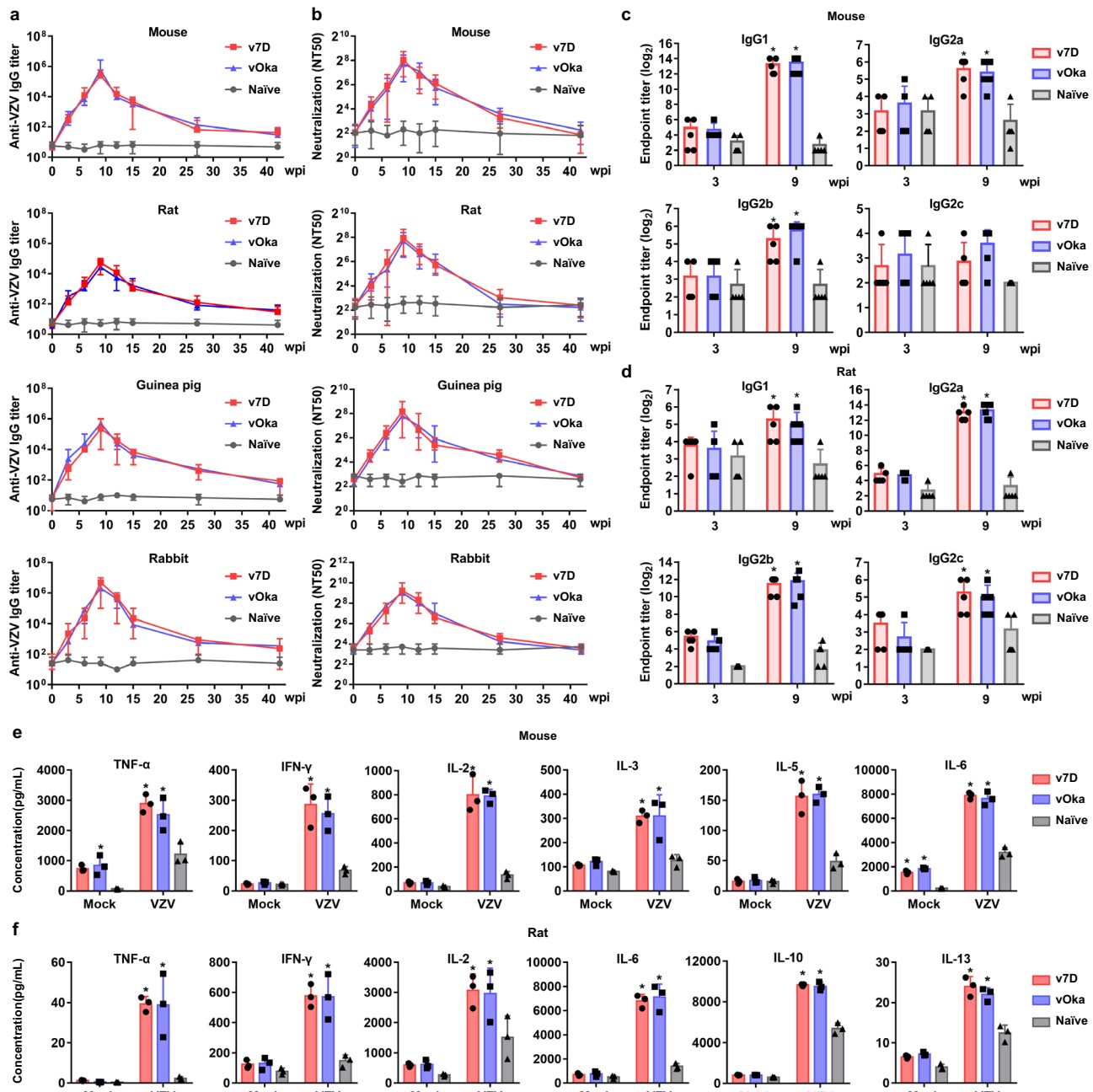

**Fig. 5 Antibody and cellular immune responses after v7D immunization in different small animals.** Mice, rats, guinea pigs and rabbits were immunized subcutaneously at weeks 0, 3, and 6 with v7D or vOka, at 500 PFU/dose for mice, 1000 PFU/dose for rats, 1000 PFU/dose for guinea pigs and 2000 PFU/dose for rabbits. **a**, **b** Serum samples were collected at indicated time points over a 42-week period for evaluation of (**a**) anti-VZV IgG antibody titers and (**b**) the neutralizing antibody titers ($n = 24$ per group for mice, $n = 18$ per group for rats and $n = 6$ per group for both guinea pigs and rabbits). **c**, **d** The IgG subclass distributions (IgG1, IgG2a, IgG2b, and IgG2c) for anti-VZV antibodies were also determined in (**c**) mice and (**d**) rats ($n = 5$ per group) 3 weeks after the first (week 3) and third inoculation (week 9). **e**, **f** Splenocytes were collected from immunized (**e**) mice and (**f**) rats ($n = 3$ per group) at week 9 and stimulated with rOka virus or mock-infected MRC-5 cell lysate as the negative control. Cytokines/chemokines (including Th1 cytokines TNF-α, IFN-γ and IL-2, and Th2 cytokines IL-3, IL-5, IL-6, IL-8, IL-10, and/or IL-13; more in Supplementary Figs. 7–8) were quantitated in the culture supernatants by Luminex at 2 dpi. The results are represented as averages ± the SD. Asterisks in (**c-f**) denote a significant difference ($P < 0.05$) compared to the untreated naïve controls as determined by two-way ANOVA with Tukey's post hoc test. Detailed information about the statistics and values are provided in Supplementary Table 3. The experiments were repeated twice, and representative results are shown. Source data are provided as a Source Data file.

stimulation, a subset of cytokines representing Th1 (TNF-α, IFN-γ, and IL-2), Th2 (IL-4, IL-5, IL-6, IL-10, IL-13, etc.) and Th17 (IL-17) responses, as well as chemokines (IP-10, MIP-1β, MCP-1, RANTES, etc.) was significantly upregulated at similar levels in splenocytes from the v7D- and vOka-immunized groups (Fig. 5e, f and Supplementary Fig. 7–8). Compared with the mock

controls, many of these cytokine/chemokine responses were virus-specific. Furthermore, the cell immune responses sustained at similar levels in both splenocytes and PBMCs from the v7D- and vOka-immunized groups until the termination of this study (week 42) (Supplementary Fig. 9–12). In addition, immunization with v7D and vOka viruses elicited comparable cellular immune

responses (characterized by TNF-α production by PBMCs) to VZV antigens in guinea pigs and rabbits (Supplementary Fig. 13). Together, these findings suggest that v7D was as immunogenic as vOka in different small animals.

**Toxicity and immunogenicity evaluation of v7D in nonhuman primates**. To further translate the v7D vaccine candidate to clinical application, the preclinical toxicity evaluations of v7D were completed in nonhuman primates. First, for neurovirulence testing of v7D (experiment schedule shown in Fig. 6a), a treatment group of macaques ($n = 10$) was inoculated once intrathalamically on both sides with a total of $3 \times 10^5$ PFU of v7D, and the control group ($n = 2$) received equal volume of normal saline. Then, macaques were monitored daily for 3 weeks. All animals survived the intrathalamic challenge. Although a decreased range of motion in the thoracic limbs was observed in four of the ten animals at 2–4 days post v7D inoculation, this also occurred in the control animals and is not likely due to treatment with v7D. This reduction in limb movement soon recovered and all animals showed no signs of central nervous system symptoms or other abnormal reactions throughout the experiment. Febrility and weight loss were not observed in any animal (Fig. 6b, c), and anti-VZV antibodies were negative in serum samples collected before and after the observation period. Following euthanasia on day 21, neurohistological analysis of different brain regions was conducted (Fig. 6d), and the needle track at or near injection sites were identified in all animals. Needle track lesions were characterized by astrogliosis, microgliosis, macrophage accumulation, etc., in all ten v7D-treated animals with nine of the animals showing mild perivascular lymphocytic cuffing near the injection site. These cellular reactions could be due to antigenic stimulation produced by the v7D inoculum or active brain infection by this vaccine virus, since astrogliosis has been reported as a hallmark of host immune responses to VZV infection in the human brain[50]. Nonetheless, there was no evidence of the expansion of neuronal necrosis and no staining for VZV gE and pORF62 that might signify virus spread into other brain regions from the injection site. Together, these results indicate that v7D lacked neurovirulence in macaques.

Second, for toxicity evaluation of repeated subcutaneous administration of v7D (experiment schedule shown in Fig. 7a), two treatment groups of macaques ($n = 10$ per group) were inoculated subcutaneously with $1 \times 10^4$ PFU (group 1) and $5 \times 10^4$ PFU (group 2) of the v7D virus, respectively, for three times at 3-week intervals. The control group ($n = 10$) received an equal volume of normal saline. After the final injection, macaques were followed up for another 6-week period. During the course of the experiments, all animals showed no signs of drug-related illness, and no death was observed. Moreover, there were no significant differences between groups in terms of body weight, body temperature, electrocardiogram or coagulation parameters, lymphocyte subsets, blood cell count, and blood biochemistry (Fig. 7b–e). Euthanasia was carried out for six animals per group three days after the final injection (day 45) and for the rest of animals at the end of the experiments (day 84). Histopathological analysis of major organs showed no evidence of systemic toxicity related to v7D administration in all animals (Fig. 7f). Further evaluation of viral tissue distribution in v7D-inoculated animals showed positive detection of VZV DNA only in the tissues at or near the injection sites but not in the tissues of major organs (Supplementary Table 2). These data indicate that repeated subcutaneous injection of v7D was tolerated well with no side effects in macaques.

In addition, during the repeated subcutaneous dose toxicity study, serum samples were collected from the macaques before and at 3-week intervals after vaccination until study termination (day 84), and anti-VZV antibody responses and serum IFN-γ levels were evaluated. The results show that v7D inoculation induced anti-VZV IgG and neutralizing antibodies in all treated animals (Fig. 7g, h). The peak GMTs of anti-VZV IgG and neutralizing antibodies for group 1 and group 2 were 6400 and 25,600, and 128 and 128, respectively. Meanwhile, serum IFN-γ levels elevated significantly to similar levels in both v7D-treated groups (Fig. 7i). Both antibody titers and serum IFN-γ levels sustained until at least day 84. These data indicated that v7D was not only well-tolerated but immunogenic in non-human primates.

## Discussion

The vOka vaccine has low virulence while inducing adequate immune responses for protection against varicella, and has been used in several countries[5,20]. However, safety concerns regarding the nervous system have been raised due to the full ability of vOka to cause cytopathogenicity, establish latency in human neurons, and reactivate to cause HZ, which poses a not-negligible risk for vaccine-associated adverse effects[6,7,12–14,20,41]. Also, the skin tropism of vOka strain contributes to the common adverse event of varicella-like rash post-immunization[7,10,11]. Given these risks, the vOka varicella vaccine is usually contra-indicated for immunocompromised individuals who are at high risk for severe varicella-related diseases[7,18]. Thus, a safer vaccine should aid in establishing worldwide herd immunity against varicella together with vOka. Herein, using vOka-induced immune signatures as a benchmark, we developed a rationally-designed skin- and neuro-dual attenuated live varicella vaccine candidate, v7D, with immunogenic properties closely similar to those of vOka but cut-down on the pathogenic concerns.

Compared with vOka, a mixture of at least eight genetically distinct haplotypes[5], v7D was a pure and genetically defined VZV vaccine strain, in which ORF7 was abrogated by a stop-codon mutation and was generated from a wild-type pOka strain. VZV ORF7 encodes a tegument protein that is conserved across human α-herpesviruses. This gene product pORF7 has the following characteristics in common with its homologs: (a) Golgi localization, (b) similar protein size (~29 kDa), (c) a conserved cysteine at position 9, a potential Golgi targeting signal, and a conserved YXXL motif at the N-terminus, (d) implicated in the secondary envelopment and virus egress, and deletion mutants produce lower titers and smaller plaques than wild-type viruses in certain cell types[31,51,52]. However, to date, it has not been reported whether deletion of UL51, the homolog of VZV ORF7, affects the skin- and neuro-tropism of other human α-herpes-viruses, including herpes simplex virus (HSV) types 1 and 2. In contrast to v7D, the UL51 deletion mutant of pseudorabies virus (PrV), a porcine α-herpesvirus, has no defect in neuroinvasion and neurovirulence in a mouse model in vivo[51]. This striking difference between VZV ORF7 and its counterpart UL51 in PrV raises the question, what specific role of ORF7 play in VZV replication? Firstly, VZV ORF7 is 65% similar and 47% identical in sequence to HSV-1 UL51 and 78% similar and 63% identical to PrV-1 UL51, which may account for differences in their ability to support virus infection of neuronal cells. Actually, VZV have a divergent requirement for other tegument proteins and glycoproteins compared to homologs in HSV and PrV[53–55], which should contribute to differences in their viral replication capacity and pathogenicity. Secondly, unique among α-herpesviruses, VZV is highly cell-associated and favors cell-to-cell spread, leading to the hallmark syncytia formation, which is crucial for VZV replication in skin and nerve tissues[56–58]. Since ORF7 is implicated in the distinctive cell-to-cell fusion process and

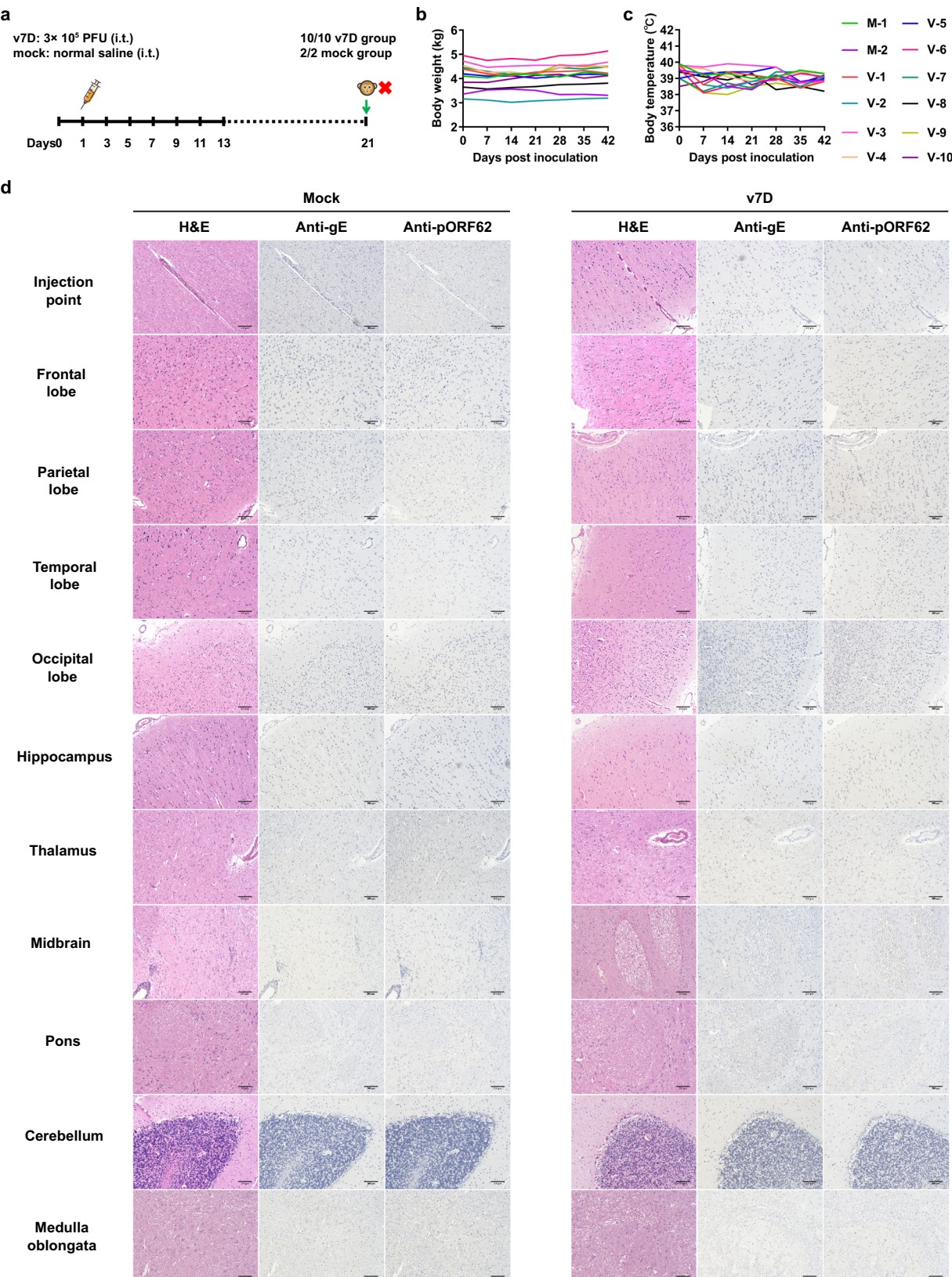

**Fig. 6 v7D lacks neurovirulence in nonhuman primates. a** Timeline of the experimental setup for the intrathalamical acute toxicity study in rhesus macaques. Ten animals were inoculated once intrathalamically (i.t.) with $3 \times 10^5$ PFU of v7D, and two animals received an equal volume of normal saline as the mock control. At 21 dpi, animals were sacrificed for histopathologic examination. **b, c** Body weights and body temperature of the animals were monitored throughout the study. M the mock group, V the v7D group. **d** Representative histopathological images of different regions of the rhesus macaque brain following intrathalamical administration of v7D at 21 dpi. The experiment was repeated twice with similar results. Scale bars indicate 200 μm. Source data are provided as a Source Data file.

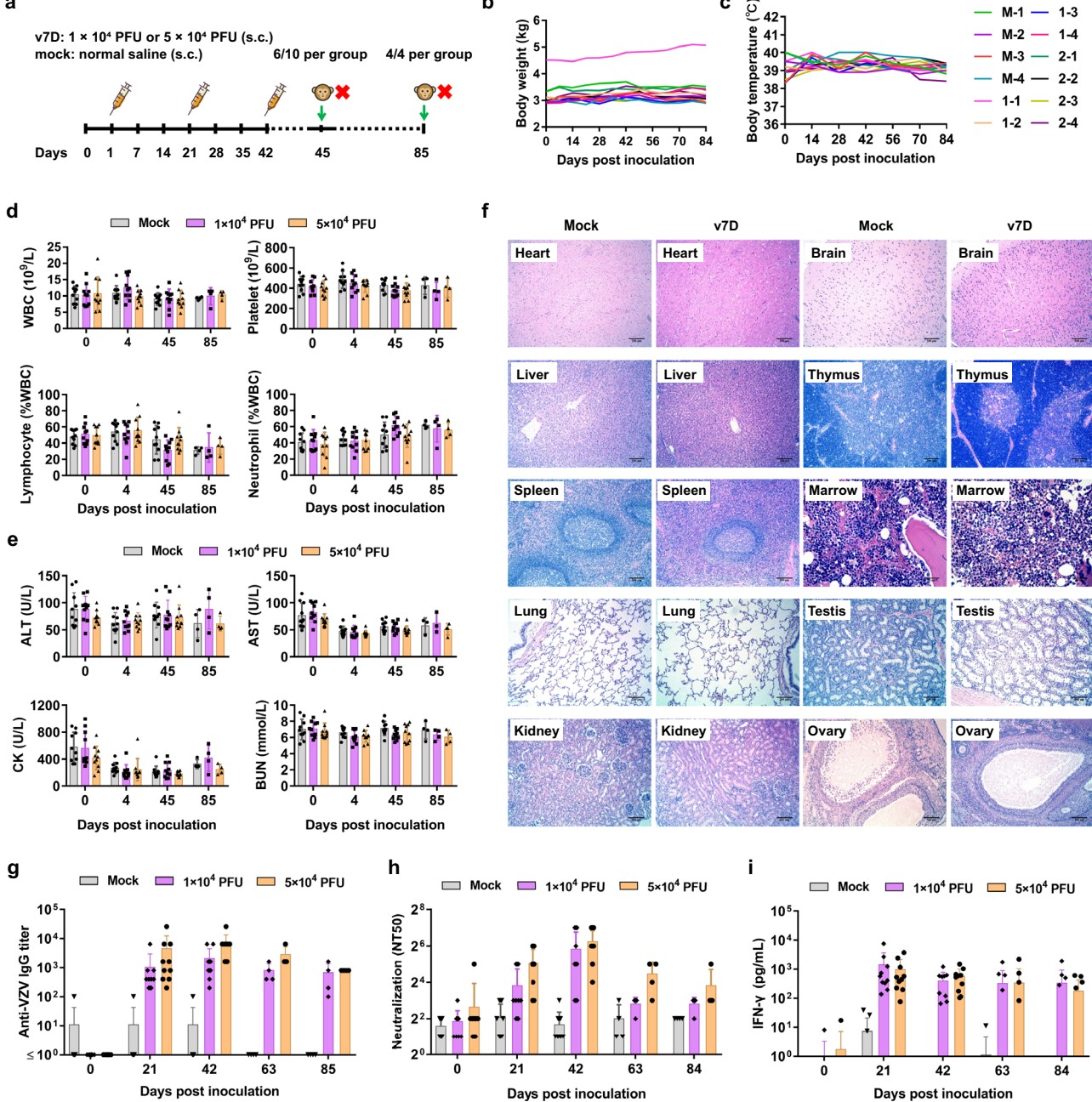

**Fig. 7 Toxicity and immunogenicity of repeated subcutaneous injections of v7D in nonhuman primates. a** Timeline of the experimental setup for the repeated subcutaneous dose toxicity study in cynomolgus macaques. Ten animals per group were inoculated subcutaneously with $1 \times 10^4$ PFU (group 1) or $5 \times 10^4$ PFU (group 2) of the v7D virus or an equal volume of normal saline three times as indicated. **b, c** Body weights and body temperature of four animals in each group were monitored throughout the study. M the mock group, V the v7D group, 1 v7D group 1, 2 v7D group 2. **d, e** Hematological parameters and serum chemistry parameters of the animals ($n = 10$ per group at day 0, 4, and 45, and $n = 4$ per group at day 85) were analyzed at the indicated time points. ALT alanine transaminase, CK creatine kinase, AST aspartate aminotransferase, BUN blood urea nitrogen. **f** Representative histopathological images of the major organs from cynomolgus macaques following repeated subcutaneous administration of $5 \times 10^4$ PFU of v7D at 45 dpi. The experiment was repeated twice with similar results. Scale bars: 200 μm or 50 μm as indicated in the respective images. **g–i** Blood samples collected from these cynomolgus macaques ($n = 10$ per group at day 0, 21, and 42, and $n = 4$ per group at day 63 and 85) were evaluated for anti-VZV IgG antibody titers and neutralizing antibody titers, and PBMCs were analyzed for IFN-γ responses in response to VZV antigens. In (**d, e**) and (**g–i**), the results are represented as averages ± the SD. Source data are provided as a Source Data file.

syncytia formation during VZV infection[31], severely impaired viral spreading of v7D within skin or nerve tissues could be due to the incapacity of the virus to fuse skin cells or fuse satellite cells with sensory neurons. Thirdly, despite having similar functions as its homologs in virion cytoplasmic envelopment, VZV ORF7 is, to our knowledge, the only one of its kind that has been

extensively evaluated in target cells/tissues of the natural host. Therefore, the observed severe defect in VZV virion packaging caused by ORF7 deficiency in human target cells/tissues should constitute one mechanism contributing to v7D attenuation and be presented as a solid evidence supporting the safety of v7D for clinical use.

Although the lack of skin- and neurovirulence of v7D is expected according to our previous work[30–32], we do not yet understand whether v7D would establish neuronal latency. To date, the latent state of VZV remains debatable[59,60]. For the SCID-hu DRG xenograft model, infection of ganglia neurons with wild-type virus or vOka was reported to be characterized by an initial lytic replication, followed by a transition to cessation of virion assembly, repression of viral gene expression and persistency of viral genomes at low levels, which was interpreted as a state of latent and/or abortive infection[41]. We obtained similar results for rOka and vOka in this study (Fig. 3). However, v7D did not establish the initial lytic infection, and thus there was an absence of not only viral gene expression (exemplified by transcription of ORF31 and ORF62 and gE expression) but also detectable levels of viral DNA in v7D-infected DRG xenografts (Fig. 3), which may suggest its severely impaired ability to establish latency in human neurons. Most recently, a VZV latency-associated transcript (VLT) that lies antisense to ORF61, together with ORF63 transcripts and a set of VLT-ORF63 fusion transcripts, has been identified in latently infected human trigeminal ganglia[61,62]. The role of these transcripts in VZV latency and reactivation has not yet been determined. However, it will be interesting to examine the expression of VLT and ORF63 transcripts when further testing v7D in in vitro models of VZV latency and in the SCID-hu DRG xenograft model.

Besides the dual-attenuation phenotype in human skin and neuronal cells, v7D grows like wild-type virus in human diploid MRC-5 cells (Fig. 1), thus permitting the production of high titers of v7D for use as live vaccines. Furthermore, v7D retains the full ability, as compared with rOka and vOka, to productively infect human immune cells, including DCs and PBMCs, in which the virus replicates its genome and viral genes are expressed without causing obvious cell damage (Figs. 2 and 4). VZV infection of human DCs and PBMCs is an essential step in virus dissemination to the internal organs and skin, facilitating virus replication within the host to cause clinical diseases[39,63]. As for the live-attenuated vaccine viruses, such as vOka and v7D, their lymphotropism that helps viral replication and dissemination could provide continuous antigenic stimulations that confer long-lasting protective immune responses against VZV infection in vaccine recipients. Meanwhile, compared to vOka, v7D showed a similar ability to functionally activate human DCs, including costimulatory molecule expression, cytokine production, antigen processing and presentation, and T-cell stimulation in vitro (Fig. 4). These processes could facilitate the priming of anti-VZV adaptive immunities and constitute a working mechanism of the v7D vaccine.

Although animal models are useful in evaluating vaccines and therapeutics against fellow α-herpesviruses like HSV and PrV, VZV is a highly human-specific virus and does not infect nor cause disease in other species[27]. Therefore, a limitation of VZV studies in general is the need to use existing animal models and in vitro systems, which lack the complexity of virus-natural host interactions, and thus only provide indirect clues as to the efficacy of v7D-induced anti-VZV immunity in humans. In this study, small animal models including mice, rats, guinea pigs and rabbits were used. Three subcutaneous immunizations of v7D elicited similar long-lasting VZV-specific antibody and cellular immune responses compared with vOka (Fig. 5 and Supplementary Figs. 7–13). Therefore, the immunogenicity of v7D was non-inferior to vOka under the same doses for both humoral and cellular immunities in these animals. The favorable outcomes from our preclinical study in nonhuman primates also strongly support the clinical translation of v7D as a live-attenuated vaccine (Figs. 6 and 7). Based on these preclinical data, v7D vaccine has advanced to a phase I clinical trial in China (ChiCTR1900022284). Compared to current vOka-based varicella vaccines, v7D vaccine is expected to demonstrate several clinical advantages through future clinical observations: (1) lower frequency of varicella-like rashes; (2) lower frequency of vaccine-strain-mediated HZ cases; (3) a possible longer protective window contributed by the higher permitted vaccine dose as a result of a deeper-attenuated vaccine strain (illustrated with a hypothetical model in Supplementary Fig. 14). However, to fully confirm the long-term safety of v7D vaccine in humans, it will require decades of observation for possible wild-type reversion, recombination events, as well as latency and reactivation of the vaccine virus, and to reveal all adverse events after vaccination.

In summary, here we report on the preclinical toxicity and immunogenicity evaluation of a skin- and neuro-attenuated live varicella vaccine v7D. Compared to vOka, v7D presents a low risk for pathogenic infection in humans and showed similar immunogenic properties in multiple animal models, inducing desired antibody and cell-mediated responses to VZV. v7D is therefore a promising vaccine candidate against varicella transmission. Further investigations on the molecular interactions of ORF7 with viral or host factors could provide more insights into its tissue-specific regulatory role in VZV pathogenesis, which could help guide the future use of v7D in prophylactic and possible therapeutic settings.

## Methods

**Construction of the recombinant viruses**. The full-length genome of the VZV wild-type pOka strain (GenBank Accession No. AB097933) was used to construct the infectious BAC clone, brOka-GFP, in our previous work[36]. This VZV BAC was maintained and propagated in *E. coli* strain SW102, and BAC modifications were carried out using galK positive/negative selection protocols[64]. Briefly, the ORF7-deleted VZV BAC was created by electroporating the PCR amplified galk expression cassette flanked with homology arms to the ORF7 gene into the VZV-BAC-containing SW102. The correct colonies were selected on minimal medium plates containing glycerol as a carbon source as well as leucine, biotin and 2-deoxy-galactose (DOG; Sigma) for selection against galK. Then, the ORF7-deficient VZV-7D BAC, b7D-GFP, was made by replacing the galk gene in the ORF7-deleted VZV BACs with a mutated ORF7 gene, in which an 11-bp region downstream of the ATG start codon was changed into a three-frame stop-codon cassette. The revertant VZV BAC, b7R-GFP, was made by rescuing the full-length, wild-type ORF7 back into the ORF7-deleted VZV BAC. BAC DNA of b7D-GFP, brOka-GFP or b7R-GFP was isolated from *E. coli* and co-transfected into MRC-5 cells using X-tremeGENE HP (Roche) with a Cre expression vector to reconstitute recombinant viruses of v7D, rOka, and 7R with a 34-bp LoxP site as the only heterologous sequence in their viral genomes (Supplementary Fig. 1). Sequences of PCR primers used for the construction of the recombinant viruses are listed in Supplementary Tables 4 and 5, and all primers were designed based upon the wild-type pOka strain.

**The Oka vaccine strain of VZV**. The Oka vaccine strain of VZV, vOka, was obtained from the American Type Culture Collection (ATCC, VR-795). The identity of this vaccine virus was confirmed by a previously established PCR–RFLP assay[43] based on specific mutations in ORF62 before use in this study.

**Cell lines and virus preparation**. MRC-5 (ATCC, CCL-171) was grown in modified minimum essential medium (MEM) with 10% fetal bovine serum (FBS) and penicillin–streptomycin (50 U/mL and 50 μg/mL) (All from Gibco). SH-SY5Y (ATCC, CRL-2266) cells were maintained in DMEM-F12 with 10% FBS and penicillin–streptomycin. To differentiate SH-SY5Y cells toward neurons (designated dSY5Y), cells were treated with 50 μM retinoic acid for 5 days, followed by treatment with 100 nM nerve growth factor (NGF) and 50 nM brain-derived neurotrophic factor (BDNF) (all from Prospec) for 7 days. Primary human dermal fibroblasts (HDFs, 2310) and human epidermal keratinocytes (HEKs, 2100) were purchased from ScienCell Research Laboratories (CA, USA) and maintained according to the supplier's recommendations.

rOka, v7D, 7R, and vOka were prepared as cell-associated viruses or a lyophilized powder of cell-free viruses. For the preparation of cell-free viruses, VZV-infected MRC-5 cells were harvested when they showed a >90% cytopathic effect (CPE). After collection, cells were resuspended in cryoprotective solution (Beijing Wantai Co., LTD) and stored at −80 °C. The cell suspension was then slowly thawed at room temperature with vigorous shaking. Next, the obtained viral samples were frozen for 2 h at −40 °C, lyophilized for 16 h at −25 °C, brought to 25 °C at a speed of 10 °C per hour, and dried at room temperature for 6 h.

The obtained lyophilized powder of cell-free viruses was dissolved in sterile phosphate-buffered saline (PBS) or normal saline and stored in aliquots at −80 °C before use. Lysates of uninfected MRC-5 cells were prepared in the same way as lyophilized powders and were used as negative controls.

**Viral growth kinetics in cell culture**. MRC-5 cells, dSY5Y cells, HDFs, and HEKs were mock-infected (isovolumetric solution of uninfected MRC-5 cell lysates) or infected in six-well cell culture plates with cell-free viruses of rOka, v7D, 7R, or vOka (MOI = 0.2 for HEKs, less susceptible to VZV infection[37]; MOI = 0.01 for the others). The cells were incubated at 4 °C for 2 h for viral entry synchronization prior to incubation at 37 °C/5% $CO_2$ for a period of 1 week. Infected cells were trypsinized daily, and titers were determined on six-well plates seeded with MRC-5 cells by plaque assay to generate growth curves.

**In vitro cell viability assay**. MRC-5 cells, dSY5Y cells, HDFs, and HEKs were seeded in 96-well plates and then infected with cell-free viruses of rOka, v7D, 7R, or vOka (MOI = 0.2 for HEKs, and MOI = 0.01 for the others). Cell viability was assessed daily over an incubation period of 1 week using a Cell Counting Kit-8 (CCK-8) assay according to the manufacturer's instructions (Beyotime Institute of Biotechnology, China).

**Infection of PBMCs**. PBMCs from healthy donors of age 20–50 years were obtained commercially from Hemacare (Van Nuys, CA, USA, PB009C). The healthy donors were screened for infectious diseases and found negative for hepatitis B virus (HBV) surface antigen, HBV core antibody, hepatitis C virus (HCV) antibody, human immunodeficiency virus −1/2 (HIV-1/2) antibody, HIV-1/HCV/HBV nucleic acid testing, and syphilis. Informed consent was obtained by the source company from all adult subjects providing PBMCs, and samples were de-identified by the company prior to our receipt. The use of human PBMCs in this study was approved by the Research Ethics Committee of Xiamen University (Approval NO. SPH-XMU2016006).

For virus infection, PBMCs were seeded in 6-well plates at a density of $1 \times 10^6$ cells per well and incubated with cell-free viruses of rOka, v7D or vOka (MOI = 0.01), or mock-infected MRC-5 cell lysate as a negative control, at 37 °C/5% $CO_2$ for a period of three days. Then, infected PBMCs were collected for the analysis of viral gene expression and genome replication.

**Infection of guinea pigs and cotton rats**. Guinea pigs and cotton rats were purchased from Shanghai SLAC Laboratory Animal Co. Ltd. and Bioray Laboratories Inc., respectively. All animals were maintained under specific pathogen-free (SPF) conditions with constant temperature (23 ± 2 °C), humidity (60 ± 10%) and 12 h dark/light cycles, and handled in accordance with standard use protocols and animal welfare regulations of Xiamen University Laboratory Animal Center. The guinea pig or cotton rat model of VZV infection was established as previously reported with certain modifications[65,66]. Briefly, groups of 7-week-old guinea pigs and cotton rats were injected intramuscularly along both sides of the thoracic and lumbar spine with infected MRC-5 cells containing $3 \times 10^5$ PFU of VZV-rOka. The animals were sacrificed one month after inoculation, and DNA/RNA was isolated from pooled thoracic and lumbar dorsal root ganglia for subsequent quantitative real-time PCR analysis.

**Infection of human xenografts in the SCID-hu mouse models**. Human fetal skin and DRG tissues (18–23 weeks gestational age) were purchased from Advance Biosciences Resources (Alameda, USA). The use of the human tissues in this study was approved by the Research Ethics Committee of Xiamen University (Approval NO. SPH-XMU2016006). Animal experiments were carried out under SPF conditions and in strict accordance with the approved animal use protocols of Xiamen University Laboratory Animal Center (Approval NO. XMULAC20160050). All surgery was performed under isoflurane anesthesia, and all efforts were made to minimize animal suffering. Animals were group housed in IVCs under SPF conditions, with constant temperature (23 ± 2 °C), humidity (60 ± 10%), and 12 h dark/light cycles.

The SCID-hu skin and DRG xenograft models were established and used as previously reported by the Arvin's lab with certain modifications[41,42]. For the SCID-hu skin mouse model, fetal skin tissues (~1 cm width × 1 cm length) were implanted on each side under the flank skin of 4- to 6-week-old male CB-17 SCID mice (Beijing Vital River Laboratory Animal Technology Co., Ltd, Beijing, China). At 3 weeks post-transplantation, the implants were surgically exposed and inoculated with ~$1 \times 10^4$ PFU (100 μL, 27-gauge needle) of rOka, v7D, 7R, or vOka. Mock-infected controls were inoculated with equal amounts of uninfected MRC-5 cell lysate. Infected-skin xenografts were harvested at 10 and 21 dpi and were then cut into pieces and homogenized with a Dounce tissue grinder (Sigma) in ice-cold MEM containing 2% FBS to release the infectious virus. After removal of the cell debris by low-speed centrifugation ($200 \times g$, 10 min, 4 °C), the obtained supernatants were used for titration by plaque assay in MRC-5 cells. The infected-skin xenografts at 21 dpi were also examined by immunohistochemistry (IHC).

For the SCID-hu DRG mouse model, human DRG tissues (~1 mm³) were implanted under the kidney capsule of SCID mice. At 4–10 weeks post-transplantation, the DRG xenografts were surgically exposed and mock-infected

or infected with ~$1 \times 10^3$ PFU (10 μL, 30-gauge needle) of rOka, v7D, 7R or vOka. Infected DRG xenografts were harvested at 14, 28, and 56 dpi and finely minced with a razor blade in ice-cold MEM for infectious virus assay and DNA/RNA extraction. The infected DRG xenografts at 14 dpi were also harvested for IHC analysis.

**Virus titration by plaque assay**. Cell-free viruses, supernatants of skin tissue homogenates, or minced DRG tissue lysates were fivefold serially diluted in MEM containing 2% FBS and 500 μL/well was inoculated onto 6-well plates with a confluent monolayer of MRC-5 cells in triplicates. After 2 h of absorption at 37 °C, the inoculum in each well was aspirated and replaced with 2 mL fresh MEM with 2% FBS. The plates were incubated for seven days at 37 °C in 5% $CO_2$, and 1 mL/well of fresh MEM with 2% FBS was added on day 3. After aspiration of the supernatants, plaques were visualized by crystal violet staining and counted under an inverted microscope. The average virus titers were determined as PFU/mL.

**Western blot analysis**. Lysates prepared from infected cells were subjected to 10% SDS–PAGE, and the proteins were transferred to nitrocellulose membranes (Whatman). After blocking in 5% skim milk in PBS for 1 h at room temperature, membranes were incubate at 37 °C for 1 h with primary mouse monoclonal antibodies of anti-pORF7 (clone 8H3[31,32], dilution 1:1000), anti-pORF9 (clone 8H6[68], dilution 1:1000), anti-pORF23 (clone 9A1, dilution 1:1000), anti-pORF40 (clone 10A2, dilution 1:1000), anti-pORF47 (clone 11H4, dilution 1:1000), anti-pORF62 (clone 1B7, dilution 1:1000), anti-pORF63 (clone 1B1, dilution 1:1000), anti-gE (clone 4A2[32,67], dilution 1:1000), anti-gB (clone 10E10[68], dilution 1:1000), anti-gH (clone 10F5, dilution 1:1000), anti-gN (clone 12E10[68], dilution 1:1000) (all made in our lab), anti-gI (Abcam, ab52552, dilution 1:1000) or anti-GAPDH (Proteintech, 60004-1-Ig, dilution 1:1000), or with primary rabbit antibodies of anti-cleaved PARP (Cell Signaling Technology, 9541 S, dilution 1:500) and anti-cleaved caspase-3 (Cell Signaling Technology, 9664S, dilution 1:500). Then, membranes were incubated with horseradish peroxidase (HRP)-conjugated goat anti-mouse (Thermo Fisher Scientific, 31430, dilution 1:5000) or anti-rabbit IgG secondary antibodies (Thermo Fisher Scientific, 31460, dilution 1:5000). Finally, protein bands were visualized by chemiluminescence (SuperSignal™ West Femto Maximum Sensitivity Substrate; Thermo Fisher Scientific).

**Immunohistochemistry**. Formalin-fixed paraffin-embedded sections (5-μm thick) of mock-infected and VZV-infected human or macaque tissues were deparaffinized and rehydrated through xylene and graded alcohols. After heat-induced antigen retrieval in 10 mM citrate buffer (pH 6.0), sections were treated with 3% hydrogen peroxide to remove endogenous peroxidase and washed in PBS. Then, sections were blocked with 10% normal goat serum and incubated with mouse monoclonal antibodies of anti-pORF62 (clone 1B7, dilution 1:5000) and/or anti-gE (clone 4A2, dilution 1:2000). Next, immunohistochemical staining was performed using an Ultrasensitive TMS-P kit (Fuzhou Maixin Biotechnology Development Co., Ltd., China) and a DAB detection kit (streptavidin-biotin; Fuzhou Maixin Biotechnology Development Co., Ltd., China) according to the manufacturer's instructions. Finally, sections were counterstained with hematoxylin, dehydrated, and coverslipped.

**Quantitative real-time DNA PCR and RNA PCR**. Infected cells or tissues were processed for extraction of DNA and RNA using the DNA/RNA Isolation Kit (Qiagen) according to the manufacturer's instructions. RNA was reverse transcribed using the PrimeScript® RT Master Mix Perfect Real-Time Kit (TaKaRa). VZV genome copies and transcript levels were determined in the presence of dually labeled internally annealing probes [6-carboxyfluorescein (FAM) and 6-carboxytetramethyl-rhodamine (TAMRA) at the 5′ and 3′ ends, respectively] for both ORF31 and ORF62 (two copies in the VZV genome) using a real-time thermocycler (Bio-Rad). Primer sequences are provided in Supplementary Tables 6 and 7. The cycling conditions consisted of a preliminary cycle at 95 °C for 30 s and 40 cycles of 95 °C for 5 s and 60 °C for 30 s. Standard curves for the quantification of DNA and cDNA amounts were generated from tenfold serial dilutions of plasmids containing ORF31 and ORF62, respectively, with detection limits of about 100 copies/mL. Samples were considered negative if no cycle threshold (Ct) value was obtained after 40 cycles in the presence of appropriate run control results.

**Generation of DCs**. CD14+ monocytes were isolated from PBMCs (Hemacare) with positive selection using Dynabeads™ FlowComp™ Human CD14 Kit according to manufacturer's instructions (Thermo Fisher Scientific). Then, monocytes were seeded in six-well plates at a density of $2 \times 10^6$ cells per well and cultured at 37 °C/5% $CO_2$ for 6 days in complete RPMI 1640 medium containing 10% FBS, 100 ng/mL GM-CSF (R&D system), and 40 ng/mL IL-4 (R&D system) to generate monocyte-derived immature DCs (iDCs). To generate v7D- or vOka-pulsed DCs, iDCs were seeded in 6-well plates at a density of $5 \times 10^5$ cells per well and incubated with cell-free viruses of v7D or vOka (MOI = 0.01), or mock-infected MRC-5 cell lysate as a negative control, at 37 °C/5% $CO_2$ over a period of 5 days. LDH release (Thermo Fisher Scientific) and cytokines/chemokine analysis (Millipore, MILLIPLEX MAP Human Cytokine/Chemokine Magnetic Bead Panel) in the supernatants during co-incubation were performed daily per the manufacturer's

instructions. GM-CSF and IL-4 were maintained in a culture medium during all DC experiments.

**In vitro stimulation of T cells by virus-pulsed DCs**. The v7D-, vOka-, or mock-cell-lysate-pulsed DCs were co-cultured with autologous CD4+ or CD8+ T cells, isolated using Dynabeads™ FlowComp™ Human CD4/CD8 Kit (Thermo Fisher Scientific) from autologous PBMCs, at a ratio of 1:10. The T-cell concentration was $2 \times 10^5$/mL. The stimulations were done in complete RPMI 1640 medium with 10% FBS over a period of 5 days. T-cell proliferation was examined daily using the 5-bromo-2′-deoxyuridine (BrdU) ELISA assay (Sigma) and by carboxyfluorescein diacetate succinimidyl ester (CFSE) labeling (Thermo Fisher Scientific) after five days of co-culture. The IFN-γ ELISPOT assay (Mabtech) was performed on day 5 after co-culture per the manufacturer's instructions. Cytokines/chemokine production in the DC-T-cell co-culture supernatants was determined after 5 days of co-culture using the Millipore MILLIPLEX MAP Human Cytokine/Chemokine Magnetic Bead Panel.

**Flow cytometry**. For analysis of VZV infection of PBMCs and DCs, cells were stained with a mouse monoclonal antibody against VZV gE (clone 4A2, dilution 1:200). For DC phenotype analysis, DCs were stained with the following antibodies: CD40 (Thermo Fisher Scientific, CD4004, dilution 1:20), CD80 (Thermo Fisher Scientific, MA1-19590, dilution 1:20), CD83 (Thermo Fisher Scientific, MHCD8304, dilution 1:20), and CD86 (Thermo Fisher Scientific, MA1-10296, dilution 1:20). Antibodies against CD4 (BD Biosciences, 561843, dilution 1:20), CD8 (BD Biosciences, 561949, dilution 1:20), and CD14 (Thermo Fisher Scientific, MHCD1401, dilution 1:1000) were used to confirm the isolation purities of monocytes and T cells or to examine DC differentiation. Irrelevant antibodies of the same IgG isotype were used to control for nonspecific antibody binding. The acquisition was performed on $>2 \times 10^4$ events using a FACSAria flow cytometer, and data were analyzed using the FACSDiva software and FlowJo software (BD Biosciences).

**Immunofluorescence assay**. PBMCs were seeded on glass coverslips pre-treated with poly-lysine and then infected with rOka, v7D, and vOka (MOI = 0.01). After incubation at 37 °C in 5% $CO_2$ for 3 days, cells were fixed in 4% paraformaldehyde for 30 min, followed by permeabilization with Triton X-100 (0.3%) for 10 min and blocked in normal goat serum. Cells were then incubated with a rabbit polyclonal antibody against pORF7 (7M[68], dilution 1:500) and a mouse monoclonal antibody against pORF62 (clone 1B7, dilution 1:2000) for 1 h, followed by incubation with both goat anti-rabbit-fluorescein isothiocyanate (FITC) (Sigma, F9887, dilution 1:500) and goat anti-mouse-tetramethylrhodamine B isothiocyanate (TRITC)-labeled (Sigma, T5393, dilution 1:200) secondary antibodies for another 30 min. Cell nuclei were counterstained with 4′, 6-diamidino-2-phenylindole, dihydrochloride (DAPI; Invitrogen), and cells on the cover glasses were observed using a Zeiss LSM 780 confocal microscope (Zeiss, Germany). Images were processed using the Zen software (Zeiss, Germany).

**Immunization in small animals**. The mice, rats, rabbits, and guinea pigs used in this study were purchased from Shanghai SLAC Laboratory Animal Co. Ltd. All animals were maintained under SPF conditions with controlled illumination (12 h dark/light cycles, 7:00 am to 7:00 pm), humidity (60 ± 10%), and temperature (23 ± 2 °C), and handled in accordance with standard use protocols and animal welfare regulations of Xiamen University Laboratory Animal Center.

The immunization route and schedule in these animals are described as follows:

(1) Groups of 4–6-week-old female BALB/c mice were immunized subcutaneously at weeks 0, 3, and 6 with 500 PFU of v7D or vOka.
(2) Groups of 6–8-week-old female Sprague-Dawley (SD) rats were immunized subcutaneously at weeks 0, 3, and 6 with 1000 PFU of v7D or vOka.
(3) Groups of 12–14-week-old female New Zealand White rabbits were immunized subcutaneously at weeks 0, 3, and 6 with 2000 PFU of v7D or vOka.
(4) Groups of 10–12-week-old female Hartley guinea pigs were immunized subcutaneously at weeks 0, 3, and 6 with 1000 PFU of v7D or vOka.

Serum samples were collected before and at 3-week intervals after vaccination until study termination (week 42) and were evaluated for antibody titers and neutralizing activities against VZV as well as IgG subclass analysis. Splenocytes and/or PBMCs were collected at week 9 (3 weeks after the final vaccination) and week 42 to determine VZV-specific cellular immune responses.

**Toxicity and preliminary immunogenicity evaluations in nonhuman primates**. Rhesus macaques and cynomolgus macaques were purchased from Beijing Prima Biotechnology Inc. (Beijing, China) and Deheng Biological Science & Technology Co., Ltd. (Nanning, China), respectively, and were maintained at the animal facility of Joinn Laboratories (China) Co. Ltd (Beijing, China). The nonhuman primate study protocol and all the experimental procedures were reviewed and approved by the local Institutional Animal Care and Use Committee (NO. ACU16-040 and ACU15-999).

For the intrathalamical acute toxicity study of v7D, 12 2.5- to 3-year-old rhesus macaques were randomly divided into two groups. A group of ten rhesus macaques, comprising five males and five females, were inoculated intrathalamically on both sides with a total of $3 \times 10^5$ PFU of v7D in 0.5 mL normal saline per animal. Another group of two animals, comprising one male and one female, were treated in the same way with the same amounts of sterile normal saline as negative controls. This procedure was performed under anesthesia with intravenous sodium pentobarbital (1 mL/kg and 30 mg/mL), and all efforts were made to minimize animal suffering. Macaques were monitored daily by facility veterinary technicians and animal caretakers for potential changes in body weight and temperature as well as clinical signs of disease (drowsiness, coma, convulsion, epilepsy, ataxia, nystagmus, hemiplegia, paralysis, chills, vomiting, or other neurological disorders) over a period of 3 weeks. Following gross anatomical observation after euthanasia, nerve tissues, including cerebral cortex frontal lobe, parietal lobe, temporal lobe, occipital lobe, hippocampus, thalamus, mesencephalon, pons, cerebellum, medulla oblongata, and spinal cord were collected and histopathologically examined by veterinary pathologists.

For the repeated-dose subcutaneous toxicity study of v7D, 30 2- to 5-year-old cynomolgus macaques were randomly divided into three groups with five male and five female animals in each group: the low-dose group (group 1), the high-dose group (group 2), and the negative control group (group 3). The low- and high-dose groups were given v7D subcutaneously at doses of $1 \times 10^4$ PFU and $5 \times 10^4$ PFU, respectively, in 2.5 mL normal saline per animal three times at 3-week intervals. The negative control group was treated with the same amounts of sterile normal saline. Toxicity was assessed by evaluating injection site reactions, clinical signs of adverse reactions, as well as changes in body weight, body temperature, electrocardiogram and ophthalmology, hematology, clinical chemistry, and urinalysis. Blood samples were also collected before and at 3-week intervals after vaccination until study termination and were analyzed for serum anti-VZV antibody responses and IFN-γ.

**Determination of IgG antibody titers**. VZV-specific antibody responses in different animal models were determined by an indirect ELISA based on VZV glycoproteins (gps) extract, namely gp-ELISA, as previously described[69] with slight modifications. Briefly, purified VZV gps were bound to 96-well polystyrene microtiter plates for 2 h at 37 °C and then blocked with a 200 μl/well of blocking buffer (2% bovine serum albumin in PBS). After washing, serial ten fold dilutions of the serum samples were prepared and 100 μL of each dilution was incubated on the VZV-gp-coated plates. HRP-conjugated goat anti-mouse total IgG (Thermo Fisher Scientific, 31430, dilution 1:5000), IgG1 (Abcam, ab98693, dilution 1:10,000), IgG2a (Abcam, ab98698, dilution 1:10,000), IgG2b (Abcam, ab97250, dilution 1:10,000) and IgG2c (Thermo Fisher Scientific, PA1-29288, dilution 1:10,000) secondary antibodies, goat anti-rat total IgG (Thermo Fisher Scientific, 31470, dilution 1:5000), IgG1 (Thermo Fisher Scientific, PA1-84708, dilution 1:10,000), IgG2a (Thermo Fisher Scientific, PA1-84709, dilution 1:10,000), IgG2b (Thermo Fisher Scientific, PA1-84710, dilution 1:10,000) and IgG2c (Thermo Fisher Scientific, PA1-84711, dilution 1:10,000) secondary antibodies, as well as goat anti-rabbit total IgG (Thermo Fisher Scientific, 31460, dilution 1:5000), goat anti-guinea pig total IgG (Abcam, ab6908, dilution 1:100,000) and goat anti-monkey total IgG (Abcam, ab112767, dilution 1:5000) secondary antibodies were used to detect specific antibody isotypes in different animal models. Endpoint antibody titers were expressed as the reciprocal of the highest serum dilution producing an OD450 greater than the cutoff value, which was defined as the average OD450 for the negative control sera.

**Plaque reduction neutralization test (PRNT)**. Heat-inactivated serum samples (at 56 °C for 30 min) were diluted twofold (1:8 to 1:4096) with MEM containing 2% FBS, mixed with equal volumes of cell-free virus suspension of VZV-rOka-GFP diluted to give ~100 PFU/well and incubated for 30 min in a 37 °C water bath. This serum–virus mixture (200 μL) was inoculated in duplicate onto MRC-5 cell monolayers in 24-well plates for 1 h at 37 °C/5% $CO_2$. Media containing viruses were replaced by 1 mL/well of fresh MEM with 2% FBS, and plates were incubated at 37 °C/5% $CO_2$ for seven days. Viral plaques with GFP expression were directly counted under an inverted fluorescence microscope (IX71, Olympus, Japan), and the highest dilutions of sera that resulted in 50% or more reduction in plaque counts were defined as the endpoint neutralization titers and expressed as their reciprocal values.

**Cytokine profile analysis in immunized small animals**. Splenocytes and/or PBMCs ($2 \times 10^6$ cells per well) from immunized animals were incubated with cell-free wild-type rOka virus (1000 PFU for mice; 2000 PFU for rats, rabbits, and guinea pigs), or with MRC-5 cell lysates as a negative control. After incubation for 2 days, the culture supernatants of splenocytes and/or PBMCs of immunized mice/rats were collected for cytokine detection using commercial multiplex bead array assays specific for mouse/rat cytokines (Millipore, MILLIPLEX MAP Mouse/Rat Cytokine/Chemokine Magnetic Bead Panel) per the manufacturer's instructions. For rabbits and guinea pigs, after incubation for 3 days, TNF-α in the culture supernatants of PBMCs was determined using commercial ELISA kits (R&D Systems) per the manufacturer's instructions.

**ELISA to detect serum IFN-γ.** PBMCs ($1 \times 10^6$ cells) from cynomolgus macaques challenged with v7D subcutaneously were incubated with 2000 PFU of cell-free rOka virus, or MRC-5 cell lysates as a negative control, along with 20 U/mL human IL-2 (R&D Systems). Half of the medium was changed every two days, and on day 5, quantitation of IFN-γ concentrations in the culture supernatants was performed with a commercial ELISA kit (Mabtech) specific for nonhuman primate IFN-γ per the manufacturer's instructions.

**Histopathological analysis.** Paraffin-embedded sections (5-μm thick) of various organs from v7D-challenged macaques were deparaffinized in xylene and rehydrated in graded alcohols. Then, sections were stained with hematoxylin and eosin (H&E), dehydrated in graded alcohol, and coverslipped. All the histopathological analysis was performed by veterinary pathologists who were blinded to treatment groups.

**Statistical analysis.** Data are presented as means ± SD, and statistical significance was determined by one-way or two-way analysis of variance (ANOVA) with Tukey's multiple comparison test at an overall significance level of 0.05 ($P < 0.05$) when three or more groups were compared to each other. Statistical analysis was conducted using GraphPad Prism version 7. Specific tests were performed as described in the figure legends.

**Reporting summary.** Further information on research design is available in the Nature Research Reporting Summary linked to this article.

## Data availability

All data associated with this study are present in the paper or the Supplementary Information. Source data are provided with this paper.

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

## Acknowledgements

This research was supported by grants from the National Natural Science Foundation of China (Nos. 81871648, 82171833, and 81601762), the National Science and Technology Major Projects for Major New Drugs Innovation and Development (No. 2018ZX09711003-005-003), and the National Science and Technology Major Project of Infectious Diseases (No. 2017ZX10304402). The funders had no role in the study design, data collection and analysis, decision to publish, or preparation of the manuscript.

## Author contributions

The project was conceptualized and designed by H.Z., T.C., X.Y., Y.L., Q.Z., T.W., J.Z., and N.X. Vaccine viruses were constructed by A.S., L.Y., J.L., J.H., J.J., and X.Y. Virological properties were characterized by W.W., D.P., W.F., Y.Z., Y.Q., C.L., and J.Y. Vaccine immunogenicity was evaluated by W.W., D.P., W.F., L.Y., R.Z., J.J., J.H., and X.Y. The manuscript was written by W.W. and T.C. and edited by J.Z., H.Z., and N.X. Data were analyzed and interpreted by all authors.

## Competing interests

The authors declare no competing interests.
