## [Peer Review File · Nature Communications]

Development of a Skin- and Neuro-Attenuated Live Vaccine for VaricellaReviewers' Comments:

Reviewer #1:

Remarks to the Author:

This extensive study describes the development of a live attenuated varicella vaccine that is attenuated for skin and neuronal infection. Their detailed studies are performed carefully addressing all the important features of varicella vaccine. Current varicella vaccine being administered to children involves the use of attenuated live varicella zoster virus (VZV). They have earlier identified VZV ORF7 as a skin and neurotropic factor using a SCID-hu model of VZV infection. They have analyzed the replication of the VZV ORF7 mutant virus (v7D) in multiple non-neuronal (including skin) and neuronal cell lines as well as human PBMCs to demonstrate successful dissemination. They also demonstrate induction of immunogenicity by v7D which is comparable to the existing vOka vaccine. Finally, they have used multiple animal models including non-human primates to show safety and immunogenicity. The manuscript is written well and their experiments are interpreted appropriately. Their work is original and very valuable in light of the complications, although rare, with the current varicella vaccine. The methods used in their experiments are valid and meets the expected standards in the field of varicella pathogenesis. Addressing the following points will additionally help the readers:

Page 7, line 143: v7D infection of MNCs was missing in the methods.

Page 10, line 199: provide a reason for using much less PFU of the virus for the xenografts.

Page 11, lines 221-231: Was the MOI of 0.01 used to avoid lysis of the DCs by VZV infection?

Page 15: line 312: they state that there was no indication of viral genetic material. But they did not analyze for VZV DNA.

Page 16; line 330: should indicate data not shown.

Page 20, line 427: they mention about an companion paper. But no attachment was found.

Page 23, line 482: Do they know the VZV sero-status of the humans whose PBMCs were used?

Reviewer #2:

Remarks to the Author:

This large group of virologists have taken up the formidable task of developing a new live attenuated varicella vaccine. The reason for this effort is stated as a need for a varicella vaccine that has less neurotropism than the current varicella vaccine (often called Oka). The Oka vaccine was developed by Professor Takahashi in the 1970s in Osaka, by a traditional approach, namely, repetitive passage of virus in cultured cells, including both guinea pig and human cells. The new approach depended on a prior survey of the properties of each of the 70 genes encoded by the VZV genome. A VZV gene that promoted neurotropism (ORF7) was deleted from the VZV genome to produce the new live attenuated vaccine called v7D. The world-wide COVID-19 epidemic has shown the scientific world that development of more than one vaccine against a human viral pathogen is a good approach. All of us have witnessed the documentation of rare but severe adverse events caused by the COVID-19 vaccines made by different companies. None of these adverse events were predictable. Only after millions of people were vaccinated did the strengths and weaknesses of individual vaccines become apparent.

Overall, the Results section is very thorough. Certainly, this group of virologists have performed more experiments than were performed in the 1970s, before the Oka vaccine was first administered to children in Japan. The experiments in which the authors injected v7D virus into the brains of monkeys are especially reassuring. A few comments are listed below. These comments are intended to improve the strength of the authors' arguments.

1. Line 59. Insert a sentence at this location to say that Professor Takahashi's group never subcloned the original Oka vaccine virus stock in the 1970's, to eliminate any variants that still possessed wild-type properties. Therefore, the current Oka vaccine is a mixture of haplotypes.

2. Line 75. Insert a new sentence and a new reference at this location. Because a relevant report has been recently published, the authors may not have seen a review of 12 cases of meningitis caused by the Oka vaccine virus in children, years after the children received their varicella vaccination. See article by E. Heusel et al, Twelve children with varicella vaccine meningitis, *VIRUSES* 12: 1078, 2020. PMID: 32292805. This report supports the authors' argument for clinical trials of a new varicella vaccine.
3. Line 158-175, Section on vivo assessment of v7D. In the text, give the exact number of guinea pigs and cotton rats that were inoculated with each of the 3 different viruses. Give exact number of animals that had R-Oka or V-Oka found in their ganglia.
4. Results, Line 260. If the ELISA was a commercial kit, give the complete name of the kit at this location in Results.
5. Results, Section on safety studies in monkeys, line 290. This section is extremely important and the authors are commended for carrying out studies in the simian brain. In lines 306-309, the authors describe finding astrogliosis and macrophage accumulation in the brains of monkeys injected with the vaccine. But the infection did not spread beyond the injection site. The authors are urged to cite a reference that shows astrogliosis is the expected pathology because both wild-type VZV and PRV cause astrogliosis after infection of the human brain (VZV) and rat brain (PRV). See article by J. Carpenter et al, *J.Virol* 90: 379, 2015 (PMID: 26491149).
6. Results, use of the word safe or safety. Lines 313-34. Abstract, line 30. Legend to Figure 6, line 1090. The word safe or the word safety may not be a good choice. Suggest that the authors say the v7D vaccine lacked neurovirulence in macaque monkeys.
7. Discussion, Lines 427-432. This paragraph needs to be modified to explain that decades will be required to reveal all adverse events after varicella vaccination. The authors are correct when they emphasize in the Introduction that neurotropism and latency following Oka vaccination are the greatest concerns about the vaccine. However, latency and reactivation will require years of observation, perhaps one or two decades of observation before the authors can state with certainty that their patients immunized in 2020 with v7D will never develop herpes zoster. Also see comment 2 and the new reference in comment 2, where vaccine viral meningitis occurred in children as late as 13 years after the first Oka varicella vaccination.
8. Discussion, Lines 399-400. That statement that VZV replicates in PBMCs without apparent effect is not correct. As shown in reference 48, cytopathic effects including viral particles on the cell surface are seen when infected PBMCs are examined by electron microscopy. Change this sentence to state that viral particles were seen in infected lymphocytes by other labs.
9. Legend to Figure 7, line 1109. Panel e is not correctly labeled.
10. Supplemental figure 14. The authors have performed an enormous number of experiments to characterize their new attenuated vaccine product. Their insertion of the raw data into 13 Supplemental figures strengthens the rigor of the research plan. However, the addition of the safety models in Supplemental Figure 14 is probably a step too far and too soon. As shown in Comments 2 and 7, the authors may need to wait 13 years to fully assess lack of neurovirulence of their vaccine in immunized children. Recommend removal of this figure.

Reviewer #3:

Remarks to the Author:

This manuscript describes a comprehensive pre-clinical evaluation of a new live-attenuated VZV strain, v7D, as a varicella vaccine. The authors present the results from cell culture, SCID-Hu mouse xenograft models of human fetal skin and DRGs, immune responses in human cells and small animal models, and safety in macaques. Overall, the manuscript is well-written and the experiments are well

done. Notably, the v7D strain is already entering human trials in China. The work presented here was likely a large part of the approval process for the human trial. These findings will be of interest to researchers in the areas of vaccines, infectious diseases, pediatrics, VZV, and more. There are some places in the manuscript where clarification is needed. It is particularly important to state clearly that the basis for the current varicella vaccine attenuation is not known, that the v7D strain was derived from the wild type parental Oka strain, and that the function of ORF7 protein is not known. These uncertainties lower enthusiasm for this strain as a new varicella vaccine, but these concerns do not override the importance of this study.

Abstract: Correct the grammar in the second sentence (line 23).

Main: Correct the grammar in the second sentence (line 40). On line 45, the word "usually" is confusing and should be deleted. This statement should be rephrased to express the concept that VZV may reactivate and cause HZ. The phrase "in some patients" is not needed because most people with HZ are not "patients" in a hospital or clinical study. On lines 66-70, the concept of vOka attenuation linked to SNPs is introduced and cited. This is a key rationale for designing the v7D strain. Since the authors are aware of the possibility that some SNPs in vOka are enriched for the ancestral (parental WT) allele in skin rashes, then it is important to explain if the v7D strain has the vOka or rOka alleles at those loci (from Ref 12: positions 107797, 105356, 106001, 19036, 97479). Finally, the consensus in the field is that no one haplotype is linked to the ability of the virus to form a rash. This idea should be addressed in the Intro and in the diagram in Supp Fig. 14. Perhaps the sequence of v7D at these key 5 loci could be included in the Supp Table 1. On line 89, the v7D strain is introduced without making it clear that it was derived from the parental Oka strain. This point is raised in the Discussion, but it should be stated earlier.

The figures in this manuscript are high quality. The consistent use of color for the different viruses is effective and helpful.

Fig. 1 and Supp Fig. 2. It is not surprising that v7D is attenuated in HEK cells. However, HEK cells did not support spread of the other VZV strains in this study. The choice of this cell line needs to be justified in the results. It is also surprising that v7D grows in MRC-5 but not HDF, which are both primary fibroblasts. This could be addressed in the Discussion. It is curious why v7D grows in lung fibroblasts but not dermal fibroblasts. Although it is stated in the Methods that the growth curves were measured by plaque assay in MRC-5 cells, it would be helpful to include the cell type used for the plaque assay in the figure caption.

Fig. 2 and Supp Fig. 3. Change "twice" to "two" in the last sentence of the figure caption.

Alternatively, remove the word "times" and keep "twice" (Twice means "two times"). It is regrettable that the DRG were not analyzed for the latency transcript, VLT. What was the level of detection of the viral load and mRNA assays? Provide this value in the caption (or on the graphs) and in the methods.

Fig. 3. If v7D does not grow in neurons, then detecting gE antigen in the infected DRGs would not be expected. If v7D established latency in DRGs, then detecting ORF31 and ORF62 mRNA would not be expected either. Furthermore, if v7D did not replicate in DRGs, then the viral genome may be below the level of detection. These points should be discussed, and the LOD for the viral load and mRNA assays provided. The presentation of the viral load data (c) and the mRNA levels (d) should be the same. A scatter plot is a better way to show this data. Furthermore, with an N=2, it is not reasonable to calculate a standard deviation. Showing both data points with a bar at the mean is better than trying to show the variation.

Supp Fig. 14. This diagram is informative and high quality. However, some statements in the caption need modification. In part b, the statement that vOka consists of both non-pathogenic and pathogenic haplotypes requires citations or should be removed. The topic of vOka attenuation, particularly which SNPs mediate attenuation and virulence, is not settled. vOka retains some skin tropism and it is not known if certain SNPs are involved. The scientific community has speculated which SNPs may be important, but this remains untested in vivo. The diagram and statement that particular haplotypes (ie "blue and purple") are pathogenic should be changed. An alternative would be that "vOka is a live attenuated vaccine that may cause varicella-like rashes..." In part c, the diagram has a bubble with the statement "Little risk of v7D latency", which should be removed. It is not known whether v7D would establish latency, although this would be expected even if the virus could not replicate in neurons. The difference in v7D is that it is much less likely to replicate and reactivate. It will be an

important research question to determine whether v7D establishes latency and expresses the VZV latency RNA transcript, VLT, in human neurons.

Discussion: It is important to explain why the v7D strain is derived from the rOka (parental) strain. Would it not be safer to make the ORF7 stop mutation in the vOka strain? This would be much easier to accept by countries that already use vOka vaccines. What is the potential for reversion of the stop codons in ORF7? Could the v7D strain recombine with WT VZV in humans? On line 429, correct the grammar in this sentence. Instead of "the less frequencies", use the term "lower frequency"

Signed: Jennifer Moffat

To facilitate your navigation, the totality of the original reviews is pasted in **black**, and our responses are in **blue**. The changes in the revised manuscript are highlighted in **red**.

=====

Reviewer Comments:

Reviewer #1 (Remarks to the Author):

This extensive study describes the development of a live attenuated varicella vaccine that is attenuated for skin and neuronal infection. Their detailed studies are performed carefully addressing all the important features of varicella vaccine. Current varicella vaccine being administered to children involves the use of attenuated live varicella zoster virus (VZV). They have earlier identified VZV ORF7 as a skin and neurotropic factor using a SCID-hu model of VZV infection. They have analyzed the replication of the VZV ORF7 mutant virus (v7D) in multiple non-neuronal (including skin) and neuronal cell lines as well as human PBMCs to demonstrate successful dissemination. They also demonstrate induction of immunogenicity by v7D which is comparable to the existing vOka vaccine. Finally, they have used multiple animal models including non-human primates to show safety and immunogenicity. The manuscript is written well and their experiments are interpreted appropriately. Their work is original and very valuable in light of the complications, although rare, with the current varicella vaccine. The methods used in their experiments are valid and meets the expected standards in the field of varicella pathogenesis.

Answer: We have addressed all the issues you raised and thank you very much for your suggestions and kind support.

Addressing the following points will additionally help the readers:

Page 7, line 143: v7D infection of MNCs was missing in the methods.

Answer: Thank you. This information is now provided in the methods, please see **lines 540-552**.

Page 10, line 199: provide a reason for using much less PFU of the virus for the xenografts.

Answer: Thanks. We have stated the tissue volumes, the gauge of the needles and the volume of viral inoculums in both the methods and results related to the SCID-hu mouse models of VZV infection (see **lines 189-191, 202-203 and lines 578, 582, 590 and 593**). This information should indicate that DRG can only be inoculated with fewer viruses because of its much smaller tissue volume (compared with that of the skin xenograft).

Page 11, lines 221-231: Was the MOI of 0.01 used to avoid lysis of the DCs by VZV infection?

Answer: Thanks. In this study, the same MOI of 0.01 was used for virus infection in most cell types (including MRC-5, SH-SY5Y, HDF, DC and PBMC) to compare in vitro growth properties of different VZV strains. Therefore, this condition was not intended to avoid DC lysis by VZV infection. Actually, human DC is resistant to VZV infection and can survive infection with cell-free VZV at higher MOIs (0.1, 1) and even incubation with VZV-infected ARPE-19 cells or fibroblasts at a 1:1 ratio for three to five days.

Page 15: line 312: they state that there was no indication of viral genetic material. But they did not analyze for VZV DNA.

Answer: Thanks. We have performed PCR analysis for the presence of VZV DNA in different tissues harvested from all v7D-inoculated macaques. Since PCR data on tissue distribution (all tissues except for the injection site) and viral shedding were negative with no Ct values obtained after 40 cycles, here we intended to simply mention these negative results and indicate data not shown (see **lines 320**).

Page 16; line 330: should indicate data not shown.

Answer: Thanks. We have made the revision as suggested, please see **line 338**.

Page 20, line 427: they mention about a companion paper. But no attachment was found.

Answer: Thanks. We co-submitted two papers regarding the preclinical and clinical studies of the v7D vaccine, respectively. The editor may have assigned different reviewers to review the two papers without adding the companion paper (clinical data) as an attachment to this one (preclinical data). This information in the text may be taken care of by the editors in the subsequent review process.

Page 23, line 482: Do they know the VZV sero-status of the humans whose PBMCs were used?

Answer: Thanks. PBMCs used in this study were obtained commercially from Hemacare Corporation. The healthy PBMC donors were screened for infectious diseases by the company, and related information has been added in the methods (please see lines 541-547). However, since VZV tests were not included in the company's routine infectious disease screening, we were unaware of the VZV serologic status of the donors.

Reviewer #2 (Remarks to the Author):

This large group of virologists have taken up the formidable task of developing a new live attenuated varicella vaccine. The reason for this effort is stated as a need for a varicella vaccine that has less neurotropism than the current varicella vaccine (often called Oka). The Oka vaccine was developed by Professor Takahashi in the 1970s in Osaka, by a traditional approach, namely, repetitive passage of virus in cultured cells, including both guinea pig and human cells. The new approach depended on a prior survey of the properties of each of the 70 genes encoded by the VZV genome. A VZV gene that promoted neurotropism (ORF7) was deleted from the VZV genome to produce the new live attenuated vaccine called v7D. The world-wide COVID-19 epidemic has shown the scientific world that development of more than one vaccine against a human viral pathogen is a good approach. All of us have witnessed the documentation of rare but severe adverse events caused by the COVID-19 vaccines made by different companies. None of these adverse events were predictable. Only after millions of people were vaccinated did the strengths and weaknesses of individual vaccines become apparent.

Overall, the Results section is very thorough. Certainly, this group of virologists have performed more experiments than were performed in the 1970s, before the Oka vaccine was first administered to children in Japan. The experiments in which the authors injected v7D virus into the brains of monkeys are especially reassuring.

Answer: Thank you for your kind support! As itemized below, we have addressed all the raised issues.

A few comments are listed below. These comments are intended to improve the strength of the authors' arguments.

1. Line 59. Insert a sentence at this location to say that Professor Takahashi's group never subcloned the original Oka vaccine virus stock in the 1970's, to eliminate any variants that still possessed wild-type properties. Therefore, the current Oka vaccine is a mixture of haplotypes.

Answer: Thanks. We have made the revision as suggested, please see lines 58-62.

2. Line 75. Insert a new sentence and a new reference at this location. Because a relevant report has been recently published, the authors may not have seen a review of 12 cases of meningitis caused by the Oka vaccine virus in children, years after the children received their varicella vaccination. See article by E. Heusel et al, Twelve children with varicella vaccine meningitis, VIRUSES 12: 1078, 2020. PMID: 32992805. This report supports the authors' argument for clinical trials of a new varicella vaccine.

Answer: Thank you very much for your kindly reminder. Varicella vaccine meningitis is a rare but important neurological complication of vaccine-associated HZ. To include this information, we have revised sentence describing consequences of vOka reactivation and inserted the mentioned reference (please see line 67).

3. Line 158-175, Section on vivo assessment of v7D. In the text, give the exact number of guinea pigs and cotton rats that were inoculated with each of the 3 different viruses. Give exact number of animals that had R-Oka or V-Oka found in their ganglia.

Answer: Thanks. Relevant information is now provided (please see line 169 and line 175).

4. Results, Line 260. If the ELISA was a commercial kit, give the complete name of the kit at this location in Results.

Answer: Thanks. Here the ELISA was not a commercial kit but our in-house established gp-ELISA, which was described in the method (please see lines 773-786).

5. Results, Section on safety studies in monkeys, line 290. This section is extremely important and the authors are commended for carrying out studies in the simian brain. In lines 306-309, the authors describe finding astrogliosis and macrophage accumulation in the brains of monkeys injected with the vaccine. But the infection did not spread beyond the injection site. The authors are urged to cite a reference that shows astrogliosis is the expected pathology because both wild-type VZV and PRV cause astrogliosis after infection of the human brain (VZV) and rat brain (PRV). See article by J. Carpenter et al, J.Virol 90: 379, 2015 (PMID: 26491149).

Answer: Thank you very much. We have made the revision to include this information and inserted the mentioned reference (please see lines 310-316).

6. Results, use of the word safe or safety. Lines 313-342. Abstract, line 30. Legend to Figure 6, line 1090. The word safe or the word safety may not be a good choice. Suggest that the authors say the v7D vaccine lacked neurovirulence in macaque monkeys.

Answer: Thanks. We agree and have changed “safe/safety” to “well-tolerated”, “toxicity” or state that v7D lacks neurovirulence in macaques (please see lines 30, 294, 296, 321, 349, 461, 735, 1125 and 1134).

7. Discussion, Lines 427-432. This paragraph needs to be modified to explain that decades will be required to reveal all adverse events after varicella vaccination. The authors are correct when they emphasize in the Introduction that neurotropism and latency following Oka vaccination are the greatest concerns about the vaccine. However, latency and reactivation will require years of observation, perhaps one or two decades of observation before the authors can state with certainty that their patients immunized in 2020 with v7D will never develop herpes zoster. Also see comment 2 and the new reference in comment 2, where vaccine viral meningitis occurred in children as late as 13 years after the first Oka varicella vaccination.

Answer: Thank you. We agree and have revised relevant descriptions, which now reads “Comparing to current vOka-based varicella vaccines, v7D vaccine is expected to demonstrate several clinical advantages through future clinical observations: 1) lower frequency of varicella-like rashes; 2) lower frequency of vaccine-strain-mediated HZ cases; 3) a possible longer protective window contributed by the higher permitted vaccine dose as a result of a deeper-attenuated vaccine strain (illustrated with a hypothetical model in Supplementary Fig. 14). However, to fully confirm long-term safety of v7D vaccine in humans, it will require decades of observation for possible wild-type reversion, recombination events, as well as latency and reactivation of the vaccine virus, and to reveal all adverse events after vaccination.” (please see lines 451-460).

8. Discussion, Lines 399-400. That statement that VZV replicates in PBMCs without apparent effect is not correct. As shown in reference 48, cytopathic effects including viral particles on the cell surface are seen when infected PBMCs are examined by electron microscopy. Change this sentence to state that viral particles were seen in infected lymphocytes by other labs.

Answer: Thanks. Here we intended to describe that infection of DCs and PBMCs with v7D, vOka or rOka, as shown in Fig. 2 and Fig. 4, did not generate obvious cell damage, thus we have changed “cytopathogenic

effects” to “cell damage” (please see line 424).

9. Legend to Figure 7, line 1109. Panel e is not correctly labeled.

Answer: Thanks. We have revised all panel labels in the legend of Fig. 7 (please see lines 1143-1149).

10. Supplemental figure 14. The authors have performed an enormous number of experiments to characterize their new attenuated vaccine product. Their insertion of the raw data into 13 Supplemental figures strengthens the rigor of the research plan. However, the addition of the safety models in Supplemental Figure 14 is probably a step too far and too soon. As shown in Comments 2 and 7, the authors may need to wait 13 years to fully assess lack of neurovirulence of their vaccine in immunized children. Recommend removal of this figure.

Answer: Thank you. We agree that decades will be required to fully assess the safety (lack of neurovirulence) and effectiveness of v7D vaccine in humans. Thus, we have revised relevant descriptions (lines 451-460) in the Discussion as suggested in comment 7. Here, we wish to preserve Supplemental Figure 14 to intuitively show the predicted clinical advantages of v7D vaccination compared with wild-type VZV infection and vOka vaccination (described in lines 451-456). We have toned down the relevant sentences in the text (see lines 456-460) and in the legend of Supplemental Figure 14 to clearly state that this is a hypothetical model showing potential safety and effectiveness of v7D vaccination in humans, which requires further confirmation by future clinical trials or real-world studies.

Reviewer #3 (Remarks to the Author):

This manuscript describes a comprehensive pre-clinical evaluation of a new live-attenuated VZV strain, v7D, as a varicella vaccine. The authors present the results from cell culture, SCID-Hu mouse xenograft models of human fetal skin and DRGs, immune responses in human cells and small animal models, and safety in macaques. Overall, the manuscript is well-written and the experiments are well done. Notably, the v7D strain is already entering human trials in China. The work presented here was likely a large part of the approval process for the human trial. These findings will be of interest to researchers in the areas of vaccines, infectious diseases, pediatrics, VZV, and more. There are some places in the manuscript where clarification is needed. It is particularly important to state clearly that the basis for the current varicella vaccine attenuation is not known, that the v7D strain was derived from the wild type parental Oka strain, and that the function of ORF7 protein is not known. These uncertainties lower enthusiasm for this strain as a new varicella vaccine, but these concerns do not override the importance of this study.

Answer: Thank you for your suggestions and kind support! As itemized below, we have addressed all the raised issues.

Abstract: Correct the grammar in the second sentence (line 23).

Answer: Thanks. The sentence has been revised, please see line 22.

Correct the grammar in the second sentence (line 40).

Answer: Thanks. The sentence has been revised, please see line 40.

On line 45, the word “usually” is confusing and should be deleted. This statement should be rephrased to express the concept that VZV may reactivate and cause HZ. The phrase “in some patients” is not needed because most people with HZ are not “patients” in a hospital or clinical study.

Answer: Thanks. The sentence has been revised as suggested, please see lines 45-46.

On lines 66-70, the concept of vOka attenuation linked to SNPs is introduced and cited. This is a key rationale for designing the v7D strain. Since the authors are aware of the possibility that some SNPs in vOka

are enriched for the ancestral (parental WT) allele in skin rashes, then it is important to explain if the v7D strain has the vOka or rOka alleles at those loci (from Ref 12: positions 107797, 105356, 106001, 19036, 97479).

Answer: Thanks. v7D is an ORF7-deficient, pOka-derived vaccine strain of VZV. The design of v7D vaccine strain is based on the identification of the skin- and neuro-tropic factor ORF7 (described in the introduction, lines 86-91) rather than SNPs associated with vOka attenuation.

Except for the mutation of the 11-bp region downstream of the ATG start codon of ORF7 into a three-frame stop codon cassette and insertion of a LOXP site in the intergenic region between ORF60 and ORF61, other regions of genomic sequence, including the mentioned loci, in v7D are the same as those in pOka or rOka. Construction of v7D is diagrammed in Supplementary Figure 1 and described in the Result and Method sections (lines 98-120 and lines 472-492).

Finally, the consensus in the field is that no one haplotype is linked to the ability of the virus to form a rash. This idea should be addressed in the Intro and in the diagram in Supp Fig. 14.

Answer: Thanks. Following your suggestion, we have added sentences in the Introduction (please see lines 71-72) and the legend of Supp. Fig. 14 to include the information that no one vOka haplotype has been confirmed responsible for vaccine-associated varicella or HZ.

Perhaps the sequence of v7D at these key 5 loci could be included in the Supp Table 1.

Answer: Thanks. Supplementary Table 1 shows the genome mutations after serial passages of v7D in cell cultures. Since the genomic sequences of the mentioned 5 loci in v7D of different passages all remained the same as those in pOka or rOka, this information was not included in this table. Besides, as mentioned above, v7D was designed based on stop-codon mutation in ORF7 rather than manipulation of any SNP linked to vOka attenuation, thus there was no analysis related to SNP in this study.

On line 89, the v7D strain is introduced without making it clear that it was derived from the parental Oka strain. This point is raised in the Discussion, but it should be stated earlier.

Answer: Thanks. The sentence has been revised to include the information as suggested (see line 92).

The figures in this manuscript are high quality. The consistent use of color for the different viruses is effective and helpful.

Answer: Thank you very much for your comment.

Fig. 1 and Supp Fig. 2. It is not surprising that v7D is attenuated in HEK cells. However, HEK cells did not support spread of the other VZV strains in this study. The choice of this cell line needs to be justified in the results. It is also surprising that v7D grows in MRC-5 but not HDF, which are both primary fibroblasts. This could be addressed in the Discussion. It is curious why v7D grows in lung fibroblasts but not dermal fibroblasts. Although it is stated in the Methods that the growth curves were measured by plaque assay in MRC-5 cells, it would be helpful to include the cell type used for the plaque assay in the figure caption.

Answer: Thanks. Keratinocytes are the main target cells of VZV infection in the skin in vivo. In recent years, the infection of VZV on primary keratinocytes in vitro has been reported in articles such as PMID: 1335482 and PMID: 24497829. Similar to our results, these studies showed that primary keratinocytes support productive VZV infection but displayed much lower virus yields and much slower rates of cell-to-cell viral spread as compared to infection of MRC-5 or MeWo cells. To justify the choice of this cell type for characterization of VZV infection in vitro, we have inserted additional references in the results (see line 135).

Indeed, it is interesting to find that v7D has a growth defect in primary skin fibroblasts but not in the lung fibroblast cell line MRC-5. This defect is similar to that observed in 7D-infected neuronal cells, such as human neuroblastoma cells SH-SY5Y (Fig. 1) and primary human neural progenitor cells (NPC) (see article by Jiang et al., PMID: 28356523). We are now doing further research to characterize the attenuation of 7D in

skin-derived cells and hope to discuss the (skin and neuronal) cell-specific role of ORF7 in VZV infection when we obtain more data in our future work.

Additionally, following your suggestion, we have included the cell type used for the plaque assay in the figure legends, please see line 1044 and line 1073.

Fig. 2 and Supp Fig. 3. Change “twice” to “two” in the last sentence of the figure caption. Alternatively, remove the word “times” and keep “twice” (Twice means “two times”).

Answer: Thanks. These have been revised, please see line 1065 and the legend of Supp. Fig. 3..

It is regrettable that the DRG were not analyzed for the latency transcript, VLT. What was the level of detection of the viral load and mRNA assays? Provide this value in the caption (or on the graphs) and in the methods.

Answer: Thanks. We are concerned about the most recent discovery of VZV latency-associated transcript, VLT (PMID: 33303747 and PMID: 29563516), which, to our knowledge, is yet to be included in the *in vivo* or *ex vivo* DRG model. We have been trying to establish relevant *in vitro* cell models and human fetal DRG models of VZV infection to analyze for VLT expression, and hope to provide new data to address whether v7D would establish latency in human neurons in our future work. Here, discussion regarding the lack of neurovirulence and the possibility of neuronal latency of v7D has been added, please see lines 400-418.

For data of the viral load and mRNA assays in Fig. 2, Fig. 3 and Supplementary Fig.3, the calculated copy numbers of VZV genome and mRNAs were normalized to GAPDH expression and the extracted total RNA, respectively. Therefore, the obtained ratio values varied depending on the quality of both the cellular and viral nucleic acid extracts. It is challenging to determine the limits of these ratio values due to a lack of standard DRG samples with known copy numbers of VZV DNA. Thus, we cannot provide the exact LOD values for the viral load and mRNA assays. Instead, the detection level of quantitative real-time PCR for viral load and mRNA quantification in this study was about 100 copies/mL, which is now provided in the methods (see line 655).

Fig. 3. If v7D does not grow in neurons, then detecting gE antigen in the infected DRGs would not be expected. If v7D established latency in DRGs, then detecting ORF31 and ORF62 mRNA would not be expected either. Furthermore, if v7D did not replicate in DRGs, then the viral genome may be below the level of detection. These points should be discussed, and the LOD for the viral load and mRNA assays provided. The presentation of the viral load data (c) and the mRNA levels (d) should be the same. A scatter plot is a better way to show this data. Furthermore, with an N=2, it is not reasonable to calculate a standard deviation. Showing both data points with a bar at the mean is better than trying to show the variation.

Answer: Thanks. Discussion regarding the lack of neurovirulence of v7D in the *in vivo* DRG model and the possibility of neuronal latency of v7D has been added, please see lines 400-418.

As mentioned above, the LOD value for the quantitative real-time PCR used in the viral load and mRNA assays is now provided in the methods (see line 655). We have revised the data presentation in panel c and d of Figure 3 as suggested, please see the revised Figure 3.

Supp Fig. 14. This diagram is informative and high quality. However, some statements in the caption need modification. In part b, the statement that vOka consists of both non-pathogenic and pathogenic haplotypes requires citations or should be removed. The topic of vOka attenuation, particularly which SNPs mediate attenuation and virulence, is not settled. vOka retains some skin tropism and it is not known if certain SNPs are involved. The scientific community has speculated which SNPs may be important, but this remains untested *in vivo*. The diagram and statement that particular haplotypes (ie “blue and purple”) are pathogenic should be changed. An alternative would be that “vOka is a live attenuated vaccine that may cause varicella-like rashes...” In part c, the diagram has a bubble with the statement “Little risk of v7D latency”, which should be removed. It is not known whether v7D would establish latency, although this would be

expected even if the virus could not replicate in neurons. The difference in v7D is that it is much less likely to replicate and reactivate. It will be an important research question to determine whether v7D establishes latency and expresses the VZV latency RNA transcript, VLT, in human neurons.

Answer: Thank you very much. Following your suggestion, the legend of Supplementary Figure 14 has been rewritten: (1) We agree that no one vOka haplotype has been confirmed responsible for the vaccine-associated diseases, and have removed any statement about the pathogenic vOka haplotype. (2) We agree that it is not known whether v7D would establish latency, and have changed the statement in the middle bubble to “v7D latency?” to indicate that the capability of v7D to establish neuronal latency remains to be elucidated (please see the revised legend of Supp. Fig. 14). Discussion regarding the lack of neurovirulence and the possibility of neuronal latency of v7D has been added in the main text (please see lines 400-418).

Discussion: It is important to explain why the v7D strain is derived from the rOka (parental) strain. Would it not be safer to make the ORF7 stop mutation in the vOka strain? This would be much easier to accept by countries that already use vOka vaccines. What is the potential for reversion of the stop codons in ORF7? Could the v7D strain recombine with WT VZV in humans?

Answer: Thanks. We agree that making the ORF7 stop mutation in a safe vOka haplotype strain rather than the wild-type pOka strain will be safer and more acceptable. However, as you mentioned, the mechanism of vOka attenuation, particularly which SNPs mediate its attenuation and residual virulence, is not settled, and no one vOka haplotype has been confirmed as a pure, safe vaccine candidate. Therefore, we did not design the 7D vaccine based on any vOka strain in this study. But we believe our research could provide insights into the further development of safe live-attenuated vaccines against varicella in the future.

Herpesviruses including VZV generally have a stable genome. Here we have shown that there was no reversion of the stop-codon mutation in ORF7 of v7D after *in vitro* passage for 25 generations (see lines 108-113 and Supplementary Table 1), which is conducive for vaccine production. However, it remains to be further investigated on possible wild-type reversion of v7D and its recombination with the wild-type virus in the human population during the long-term safety study of v7D vaccination. Discussion on these concerns, together with the concern on neuronal latency of v7D, has been added in lines 456-460.

On line 429, correct the grammar in this sentence. Instead of “the less frequencies”, use the term “lower frequency”

Answer: Thanks. The sentence has been revised as suggested (please see line 453).

Reviewers' Comments:

Reviewer #1:

Remarks to the Author:

The authors have satisfactorily responded to the queries raised by this reviewer.

Reviewer #2:

Remarks to the Author:

The varicella vaccine results are very noteworthy.

The work is significant to the varicella vaccine field

The conclusions are justified by the presented data.

The authors have corrected any flaws in their revision.

The authors have presented more than enough data to verify their conclusions,

Well done.

Reviewer #3:

Remarks to the Author:

This revised manuscript is substantially improved. The authors responded appropriately to my comments. In particular, the changes to the Supp Fig. 14 were very good. The new diagram and legend makes it clear that the attenuation of vOka is not defined. The issue of providing the Level of Detection (LOD) for the viral load and mRNA PCR assays was addressed adequately.